# A political economy theory of fossil fuel subsidy reforms in OECD countries

Nils Droste [1] ✉, Benjamin Chatterton [2] & Jakob Skovgaard [3]

Fossil fuel subsidies continue to be a considerable barrier to meeting the targets of the Paris Agreement. It is thus crucial to understand the political economy of fossil fuel subsidies and their reform. To understand these mechanisms in the developed world, we use a database of different types of fossil fuel subsidy reforms among Organisation for Economic Co-operation and Development (OECD) countries. We find evidence for four intertwined processes i) a market-power mechanism: higher market shares for renewables ease fossil fuel subsidy reforms, and ii) a policy mechanism: reforms reduce the levels of fossil fuel subsidies. Importantly, both effects are contingent on iii) a polity mechanism where institutional quality influences the feasibility and effectiveness of political reforms, and iv) a feedback mechanism where systemic lock-ins determine the effectiveness of market competition. Our results even suggest that reforms carried out by effective governments with low corruption control are associated with increasing subsidies per capita. Renewable energy support can however provide a leverage point to break path-dependencies in fossil fuel-based economies. This turns out to be more effective when coupled with improvements to institutional quality and the insulation of political processes from pro-subsidy interests.

Subsidies for the consumption and production of fossil fuels constitute crucial, yet under-theorized barriers for meeting the Paris Agreement targets. Fossil fuel subsidies including tax breaks, production subsidies and capped fuel prices are estimated at 300–1000 billion dollars per annum[1–4]. The Intergovernmental Panel on Climate Change (IPCC) estimates that removing fossil fuel subsidies can reduce global emissions by 1–10 percent by 2030, not including the effects from disrupting lock-in of fossil fuel production and consumption patterns[2,5–9]. Furthermore, there are many obvious benefits to removing fossil fuel subsidies beyond the positive climate impact: the fiscal benefits of phasing out large subsidies, the distributional benefits of removing regressive subsidies, and the health benefits from the reduced use of fossil fuels[10–12]. Despite international commitments to reform and phase out inefficient fossil fuel subsidies within the Group of 20 (G20), and at the Glasgow and Sharm el-Sheik Climate Conferences, their levels declined to a limited degree[1,13] before rising to an all-time high in the energy crisis in 2022[14]. Given these political

commitments, and a fair level of acceptability in constituencies[15], the persistence fossil fuel subsidies is a puzzling lock-in[16,17] and motivates the subsequent analysis.

When it comes to the existing literature, recent studies of fossil fuel subsidies and their reform have mainly focused on subsidies in countries with medium to low human development index (HDI) scores (as defined by the United Nations Development Program), especially subsidies that consist of fixing the price of petroleum products at below market rates[18–24]. The existing literature has thus already identified several factors that may influence fossil fuel subsidies and their reform. Whether similar mechanisms apply in high and very high HDI score countries where production subsidies and tax reductions for fossils fuels dominate is an open question. This is particularly important if high income countries are to play a leadership role and remove fossil fuel subsidies at a rapid pace[11]. Most importantly however, the degree to which fossil fuel subsidy reforms reduce fossil fuel subsidy levels remains largely unexplored. Here,

[1]Department of Political Science and Centre for Innovation Research, Lund University, Box 117, 221 00 Lund, Sweden. [2]Department of Economic History, Lund University, Box 117, 221 00 Lund, Sweden. [3]Department of Political Science, Lund University, Box 117, 221 00 Lund, Sweden. ✉e-mail: nils.droste@svet.lu.se

we shortly introduce what is already known and where the gaps in our knowledge are.

First, fossil fuel subsidies constitute interventions in the energy sector, and often have as their declared objective to enhance economic growth, support low-income groups, or promote specific sectors. Scholars have identified the size of fossil fuel reserves and global fossil fuel prices as important drivers of fossil fuel subsidies[25–28]. Other economic factors such as the level of income also play a role regarding fuel price subsidies[13]. Furthermore, fossil fuel prices are important as higher prices increase demand for consumer subsidies but also increase the fiscal costs of supplying them and thereby create government incentives to reform such subsidies[29–33]. Finally, regarding the competition between different sectors, companies in the renewable energy sector can benefit from environmental regulations, subsidies and taxes, and may be hurt by fossil fuel subsidies, whereas fossil fuel intensive companies follow the inverted pattern[34]. Policy interventions supporting renewable energy helps to transition the economy away from fossil fuel[35,36]. Taken together, these factors point to wider sets of economic and energy-related factors that evolve around competition and governance of technologies in the energy market. What is less studied empirically is the argument that competition in the market, measured by the share of renewable energy, supports the reform of fossil fuel subsidies, e.g. by disrupting fossil fuel lock-in[36].

Second, the existing fossil fuel subsidy literature has identified the demand for subsidies from special interest groups, both producer and consumer, as a factor[18,20,23,37]. Other, related studies highlight the role of organized special interests in hindering climate policy more generally and how these groups may be enabled or hindered by the political-institutional context[38–40]. Subsidies provide concentrated benefits for specific interest groups that produce or consume fossil fuels, subseqently increasing their share of the economy, and hence their power[18,41,42]. Similar dynamics emerge regarding trade unions covering affected sectors (e.g. the coal industry), politicians with ties to sectors or regions that receive subsidies, and to state-owned fossil fuel companies[20,21,23,43]. Other groups may seek to remove subsidies, fiscal actors such as finance ministries have an interest in reducing expenditure[44], and environmental groups to lower fossil fuel usage[45]. Consequently, pro-subsidy coalitions may form across different industry sectors and policy-makers, and anti-subsidy coalitions across industry sectors (e.g. the renewable energy sector) and environmental and fiscal actors[46]. This points to the importance of interest groups and their ability to influence political decisions in their own interest. Only few studies have looked into the role of corruption as a way for interest groups into influence fossil fuel subsidies in developing countries[47,48]. It is thus less clear, how corruption and institutional quality in general influences fossil fuel subsidies and their reform in industrialized countries. This is important since institutional quality – unlike pro- and anti-subsidy coalitions and energy factors – generally differ between industrialized and developing countries.

Third, fossil fuel subsidy scholars have also recognized the importance more structural political factors, including the kind of political regime[24,25,49]. These include public involvement in the energy sector such as state-owned oil enterprises[22], institutional or governance capacity[22,50], and the social contract between the state and citizens[51,52]. Such structural factors lock-in the current system by hampering change through path dependencies[19,23]. For instance, fossil fuel subsidies may entrench behaviors that are fossil fuel dependent, e.g. using more fossil fuel-intensive vehicles or machinery, which may galvanize the actors engaging in this behavior to oppose subsidy reform[53]. Therefore, scholars have argued that the context and conditions for reforming fossil fuel subsidies and transitioning away from fossil fuels are not politically neutral, but favor fossil fuel incumbents and the status quo[17,42,53–55]. This points to system dynamics that keep the state of the system constant through self-reinforcing feedback-loops.

This article introduces a dataset mapping out fossil fuel subsidy reforms from across 35 Organisation for Economic Co-operation and Development (OECD) countries from 2005 to 2020 (see Fig. 1 for an overview). These countries have so far been under researched due to a focus on the developing world[27,48,56,57]. The dataset categorizes fossil fuels subsidy reforms by type and covers subsidy initiatives ranging from tax expenditures and direct budgetary transfers to royalty exemptions and price fixing (see Methods for details). We use this dataset to abductively develop a theory on the political economy of fossil fuel subsidy reforms. We first inductively identify renewable energy share and institutional quality as important factors in explaining the persistence of fossil fuel subsidies through regression trees. From there, we build a political-economy theory on the mechanisms that can explain both the persistence of fossil fuel subsidies and highlight potential leverage points to remove lock-in into such subsidies. Our political economy approach shall be understood as one that studies of the relationship between economy and politics. Specifically, we draw on the emerging field of the political economy of energy, and its core tenet that "energy systems are inherently political and economic as much as they are technical or technological"[58]. This approach often involves studying political economy factors such as the role of special interests, energy (especially fossil fuel) resources, levels of income, and political institutions. Fossil fuel subsidies fit well into such an approach, as they constitute political interventions in energy markets, but also as they constitute a key components of the energy system and are influenced by markets actors' demands for subsidies[59]. Here, we contribute in two ways: empirically, by providing data on fossil fuel subsidy reforms in OECD countries, and theoretically by building and testing a theory on the mechanisms that lead to reforms of fossil fuel subsidies. We provide empirical evidence for the explanatory power of our theory by showing that i) larger renewable energy shares decrease the role of fossil fuels and thereby ease the reform of fossil fuel subsidies, and ii) reforms lead to lower overall fossil fuel subsidy levels. We also find evidence of iii) feedback loops that makes it harder to remove fossil fuel subsidies the higher the total level of subsidies in the country. Yet, our main contribution is to show how institutional quality affect the reform outcomes. High levels of government effectiveness generally lead to greater numbers of reforms and thus lower subsidies. However, high levels of government effectiveness coupled with low corruption control tends to lead to increases in subsidies per capita as more reforms are passed, indicating regulatory capture of the reform process. These results indicate two ways out of fossil fuel lock-ins. A long-term solution would thus be to improve the quality and functioning of political institutions, in particular insulating the political process from vested interests. The more immediate solution is to support alternative technologies such as renewables that make the overall socio-economic system less dependent on fossil fuels and increase the likelihood of reforms.We thereby show a way out of political and economic lock-in effects in countries with per capita $CO_2$ emissions that are substantially above the global average, and with corresponding responsibilities[11]. We thus contribute to the wider literature on fossil fuel subsidies and the role of institutions in climate politics[17,36,38,53,60].

## Results

We employed an abductive approach, in which i) the initial inductive identification of relevant variables is used to ii) develop a theory about causal mechanisms rooted in and advancing existing literature and iii) subsequently test a set of hypotheses derived from the theory. Our initial investigation highlighted factors consistently contributing to the likelihood of reform, with weaker explanatory power attributed to other factors such as oil prices, state owned fossil fuel companies, and high levels of fossil fuel production. For detail on the inductive theory

# Fossil fuel subsidy reforms and CO2 emission per capita

**Fig. 1 | The distribution of fossil fuel subsidy reforms and CO$_2$ emissions per capita in countries within the Organisation for Economic Co-operation and Development.** The bivariate choropleth map shows the distribution of average fossil fuel subsidy reforms and average CO$_2$ emissions per capita from 2005 to 2020 in terciles among both dimensions.

building see Supplementary Information. Here, we present a short outline of our political economy theory on fossil fuel subsidy reforms (see Fig. 2).

This theory was developed by an inductive approach to data analysis as well as deductively placing the above-mentioned factors in a wider theoretical context. Through this overall abductive method, we stipulate and test the corresponding causal mechanisms. Below in the sections on the different mechanisms we elaborate the theoretical backgrounds for each of them. A more overarching theoretical foundation consists of our political economy of energy-approach, which seeks to explain energy politics in terms of a set of actors with varying objectives that seek to influence policy-making within the context of political institutions[61]. In the case of energy policy in general, the relative power of fossil fuel and renewable energy constituencies – who stand to win or lose from green energy transitions as well as fossil fuel subsidies – are particularly relevant[39,62]. Their relative power is inter alia shaped by domestic political institutions such as the level and kind of democracy and the channels of access to policy-making of these coalitions[24,38,55].

Specifically, our theory consists of four interconnected mechanisms: i) a market-power mechanism that captures the effects of competition between energy sectors, and where larger renewable energy shares decrease the role of fossil fuels and thereby ease the reform of fossil fuel subsidies (see the causal arrow from renewables' market share to fossil fuel subsidy reform in Fig. 2); ii) a policy mechanism where reforms lead to lower overall fossil fuel subsidy levels (see the causal arrow from fossil fuel subsidy reform to fossil fuel subsidy levels in Fig. 2); iii) a polity mechanism that captures the effects of institutional quality on the effectiveness of the market and policy mechanisms (e.g. government effectiveness, corruption control) (see the moderating causal arrow from quality of institutions to the causal arrows of mechanisms i and ii in Fig. 2); and iv) a feedback mechanism where lock-ins into fossil fuel dependent pathways affect the

effectiveness of the market mechanism (see the causal arrow back from fossil fuel subsidy levels to renewable market share). The mechanisms in our theory include energy markets, political and institutional (feedback) factors, similarly to other theoretical frameworks that include techno-economic, socio-technical and political factors to study e.g. energy transitions[63]. Next, we introduce these mechanisms, present the corresponding causal hypothesis inspired by both our inductive approach to the data and our deductive engagement with the literature. To test the developed hypotheses, we present results of two-way fixed effects regressions that measure within country variations[64] and account for moderating effects[65] of mechanisms iii and iv (see Methods).

## A market-power mechanism of technologies and interests

We theorize that market share influences the power of interest groups. The higher the share of renewables in the energy market, the stronger their industry associations, and the lower the legitimacy of fossil fuel subsidies[36]. Because it is in their interest to phase out subsidies that support competing technologies, an uptake of renewable technology increases the likelihood of reform to fossil fuel subsidies. For instance, as renewable energy becomes a competitive alternative to fossil fuels the power of anti-subsidy coalitions relative to pro-subsidy coalitions may increase[66,67]. Renewable energy may break the carbon lock-in of the fossil fuels, which is inter alia upheld by fossil fuel subsidies[68]. Therefore, we postulate hypothesis 1 (H1, Fig. 2): The larger the market share of renewable energy, the easier it is to reform fossil fuel subsidies.

Figure 3 shows that for 35 OECD countries from 2010 to 2019 there is a significant, positive, within-country relation of renewable energy consumption shares and the number of reforms based on a two-way fixed effect regression including covariates. We thus find evidence that renewable energy shares increase the likelihood for fossil fuel subsidy reforms.

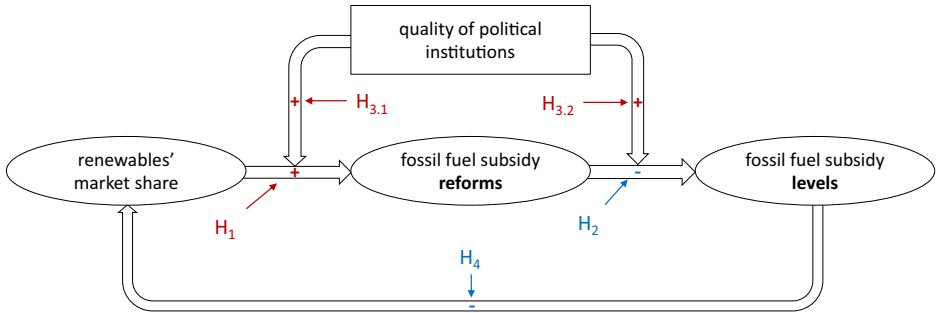

**Fig. 2 | A theory of fossil fuel subsidy reforms.** The structural causal model shows how energy market and political institutions interact when it comes to fossil fuel subsidy reforms. The ovals represent variables, the hypothesized causal relation are symbolized by arrows, and the box represents a concept that we operationalize with multiple variables (see Methods). Red indicates a positive hypothesized relation and blue a negative one.

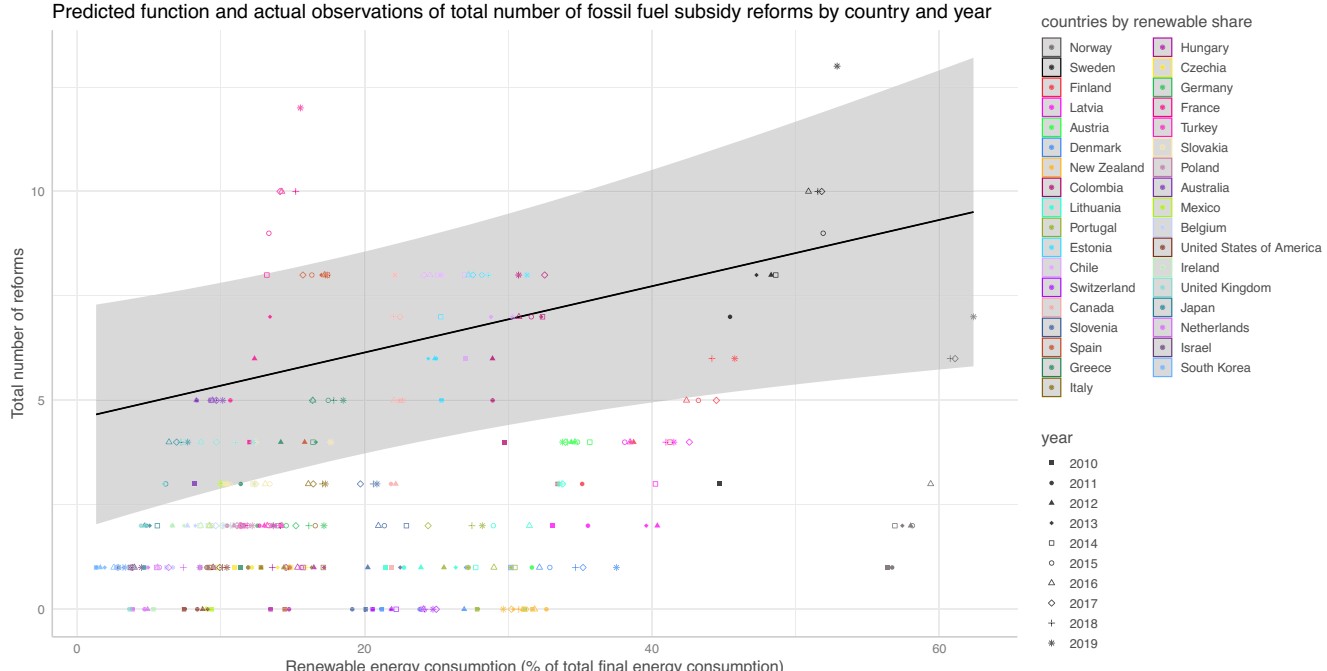

**Fig. 3 | Evidence for the market-power mechanism.** The figure shows the correlation between renewable energy shares in the energy market and the number of reforms. In addition to the estimated functional relation, the plot shows country and year-based observations. Confidence intervals are gray shaded areas.

## A policy mechanism of effective reforms

We hypothesize that, in principle, fossil subsidy reform has a measurable effect on the overall levels of subsidies. This is not necessarily the case, as reform of one subsidy may be reversed or offset by increases in other fossil fuel subsidies[69]. The mechanism here is that reforms of fossil fuel subsidies should lower overall fossil fuel subsidy levels. Correspondingly, we hypothesize: Reforming fossil fuel subsidies reduces the level of subsidies (H2, Fig. 2).

Figure 4 shows that for 35 OECD countries from 2010 to 2019 there is a significant, negative, within-country relation of the number of reforms and the total level of fossil fuel subsidies per capita, based on a two-way fixed effect regression including covariates. We thus find evidence that fossil fuel subsidy reforms on average lower subsidy levels.

## A polity mechanism of institutional quality

The third type of mechanism is placed on the system level and captures how structural aspects of the socio-economic system influence the functioning of the main causal chain through the market and policy mechanisms. We theorize that institutional quality influences how well the market-power and policy mechanisms

function, and operationalize institutional quality at various stages in terms of electoral democracy, government effectiveness and corruption control. We expect that a higher quality of political institutions decreases the possibility of pro-fossil fuel subsidy coalitions preventing subsidy reform. Political institutions including the level of democracy have been shown to affect fossil fuel subsidies[20,24]. Here we utilize multiple variables as a proxy of institutional quality but finally select the electoral democracy index as the most power full explanatory variable to represent institutional quality (see Method for details on the operationalization). Existing studies of the impact of electoral democracy on fossil fuel subsidies have provided slightly contradicting results regarding whether there is such an effect[22,70]. On this background, we find it worthwhile to explore whether higher institutional quality increases the number of reforms. Our corresponding hypothesis is: The positive correlation between renewable energy share and fossil fuel subsidy reforms is positively influenced by electoral democracy (H3.1, Fig. 2). We estimate this through an interaction effect of electoral democracy index with the share of renewable energy consumption.

Figure 5 shows that irrespective of high or low electoral democracy values there always is a positive relation between renewable

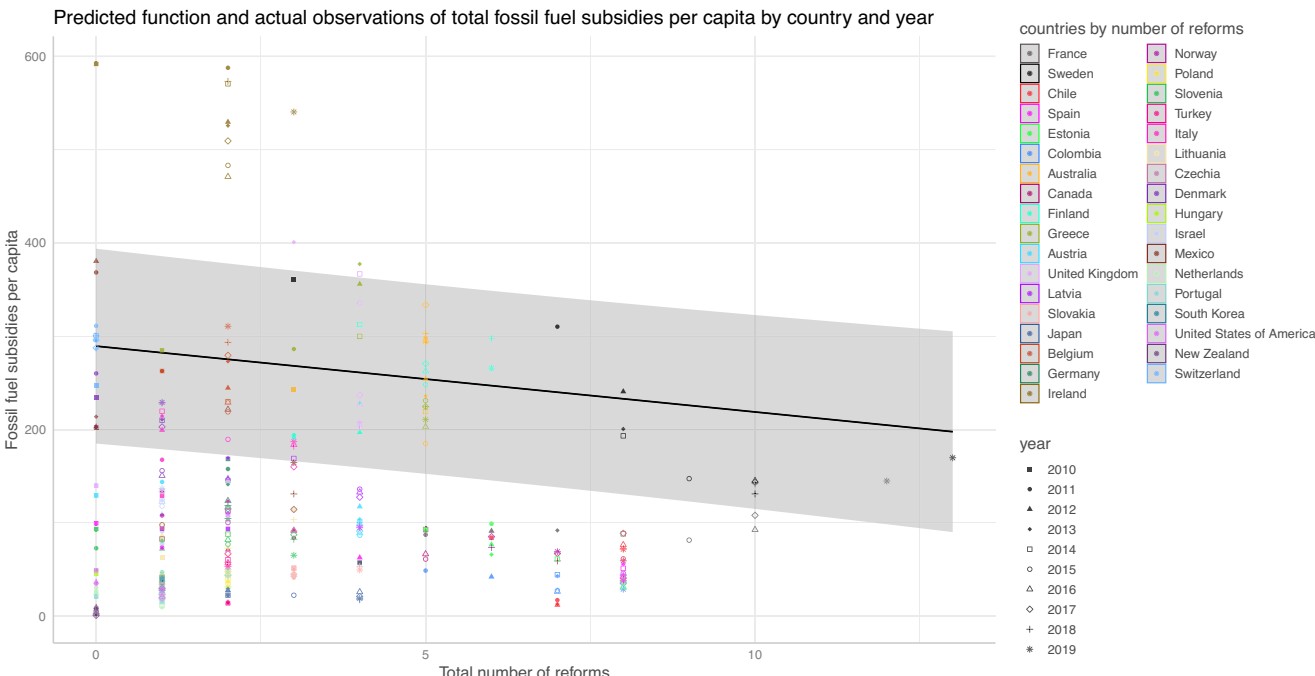

**Fig. 4 | Evidence for the policy mechanism.** The figure shows the correlation number of reforms and the level of subsidies per capita. In addition to the estimated functional relation, the plot shows country and year-based observations. Confidence intervals are gray shaded areas.

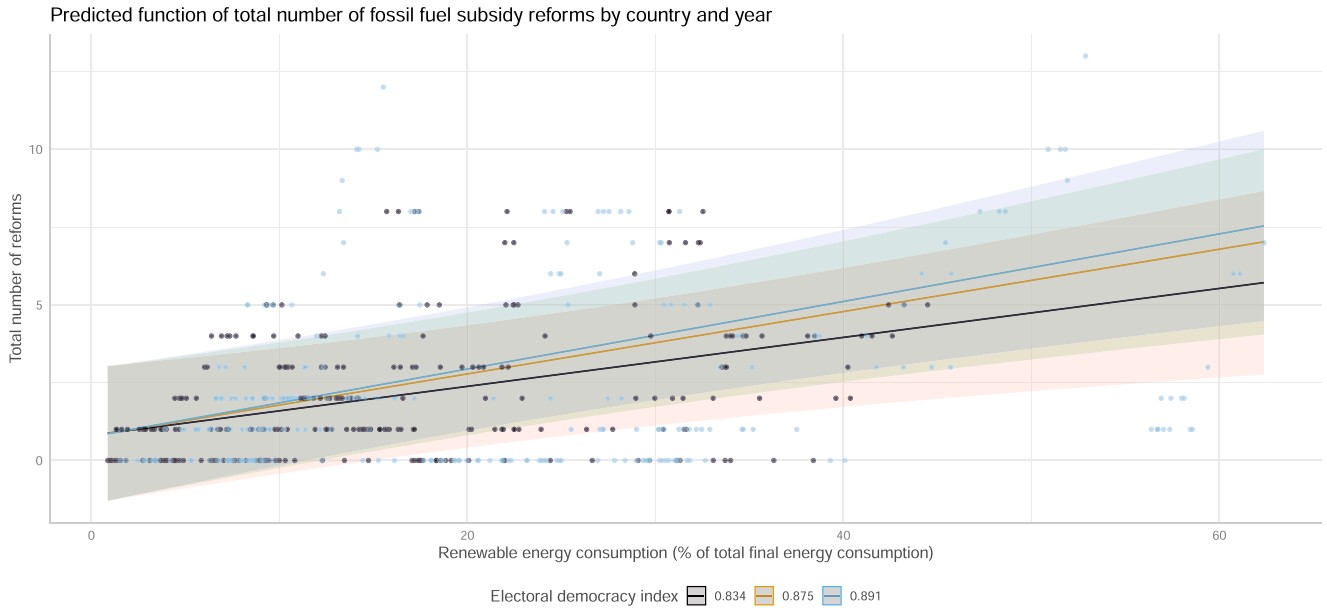

**Fig. 5 | Evidence for the polity mechanism.** The figure shows the correlation between renewable energy shares in the energy market and the number of reforms. The model estimates how electoral democracy moderates this relation. To indicate the interaction effects between renewable energy and electoral democracy, the plot shows the estimated functional relation at three different values of electoral democracy (with black indicating a value of 25% percentile, yellow indicating the median, and blue indicating 75% percentiles). Raw observations are colored by black for below median electoral democracy index and blue for above median electoral democracy index values. Confidence intervals are shaded areas.

energy consumption and number of reforms. However, the corresponding interaction effect is significant and the effect of renewable energy shares on total number of reforms is larger for countries with a high electoral democracy index. Because the index is composed by indicators of electoral freedom and voting rights, it represents government responsiveness through free electoral competition. We thus find evidence that electoral democracy increases the functioning of market competition between alternative energy sources such as renewable energy and fossil fuels.

Furthermore, we hypothesize that institutional quality also affects the policy mechanism. In short, the better political institutions function, the larger the effects of reforms. Previous studies have however also found that corruption has an effect on fossil fuel subsidies, at least in developing countries[19]. Thus we broaden our operationalization of institutional quality to include not only government effectiveness but also to which degree corruption is limited, what is referred to as corruption control (see Methods). Thus, we formulate hypothesis 3.2: The negative correlation between fossil fuel subsidy reforms and fossil fuel

subsidy levels is positively influenced by institutional quality in terms of government effectiveness and corruption control (H3.2, Fig. 2).

Figure 6 shows the relationship between institutional qualities and the effects of reform. On average subsidies decrease with additional reforms. However, we find mixed evidence for H3.2. We find that the number of reforms is positively influenced by government effectiveness as the corresponding two-way interaction is significant (see Methods). However, the two-way interaction with corruption control is not significant. We again allowed for a three-way interaction to capture further potential heterogeneity. Importantly, the positive effect of government effectiveness is conditional on corruption control. The corresponding three-way interaction effect is significant (see Methods for detail). Reforms in effective but corrupt governments thus seem to work differently from reforms in effective but less corrupt governments. Figure 6 displays that it is important to differentiate. In situations with low corruption control (Fig. 6a), a high level of government effectiveness (blue line) leads to higher subsidies following reforms. This implies that interest groups in favor of fossil fuel subsidies effectively benefit from corrupt but effective governments when reforms are undertaken. In situations with high corruption control (Fig. 6c), government effectiveness makes little difference, and more reforms always reduce the level of fossil fuel subsidies.

### A feedback mechanism of entrenching lock-ins
The second systemic effect is based on a theory of lock-in effects that entrench the system in its current state. Here, we theorize that the more entrenched vested interests are, i.e. the more the socio-economic system is based on fossil fuel subsidies, the less effective the other mechanisms, and the more difficult it is to reform. We do base this on the basis of existing literature on the lock-in or entrenchment of fossil fuels, and the role of fossil fuel subsidies in this lock-in[42,68]. In particular, we assume that the higher the existing subsidies, the harder market entry for renewables. Thus, we hypothesize: The higher fossil fuel subsidy levels, the lower the renewable energy market share (H4, Fig. 2).

Figure 7 shows that the subsidy levels reduce renewable energy shares, although to a small degree. The corresponding coefficients (−0.0025) is significant at 10% level. Therefore, we find some kind of support for the notion that the total level of subsidies per capita constitute part of a lock-in. Conversely, it also means that the less fossil fuel subsidies in a political system, the slightly easier it is for renewables to gain market share – which in turn makes reforms more likely. The feedback loop can go both ways, entrenching or transformative.

## Discussion
Developing and testing a political economy theory of fossil fuel subsidy reform on a dataset of OECD countries provided a range of insights. First, we postulate a market-power mechanism where the share of renewable energy increases the likelihood of reform. Second, fossil fuel subsidy reforms on average lead to lower subsidy levels. Through two-way fixed effects regression analyses we provide evidence for both mechanisms across 35 countries and 10 years of observations. Our main contribution lies in identifying structural mechanisms that further support or hinder change in industrialized countries. We hypothesized that institutional quality support both the market-power mechanism and the policy mechanism. Here, we provide quantitative evidence that institutional quality can substantially ease fossil fuel subsidy reform, in line with previous findings[19,47]. In addition, in line with previously theorized mechanisms of fossil fuel subsidy lock-in[42], we hypothesized that higher fossil fuel subsidy levels deter the market entry of renewable energy and contribute supporting empirical evidence. However, our results go further however and suggest that institutional quality not only eases the passage of reforms, but also governs their effect on subsidy levels. Through an interaction model, we provide a differentiated picture. In cases with high corruption control we find that additional reforms generally lead to lower subsidy levels. However, under conditions of weak corruption control, effective governments tend to increase fossil fuel subsidies with additional reforms that are officially intended to remove the subsidies. This indicates capture by vested interests. These findings show important similarities with the

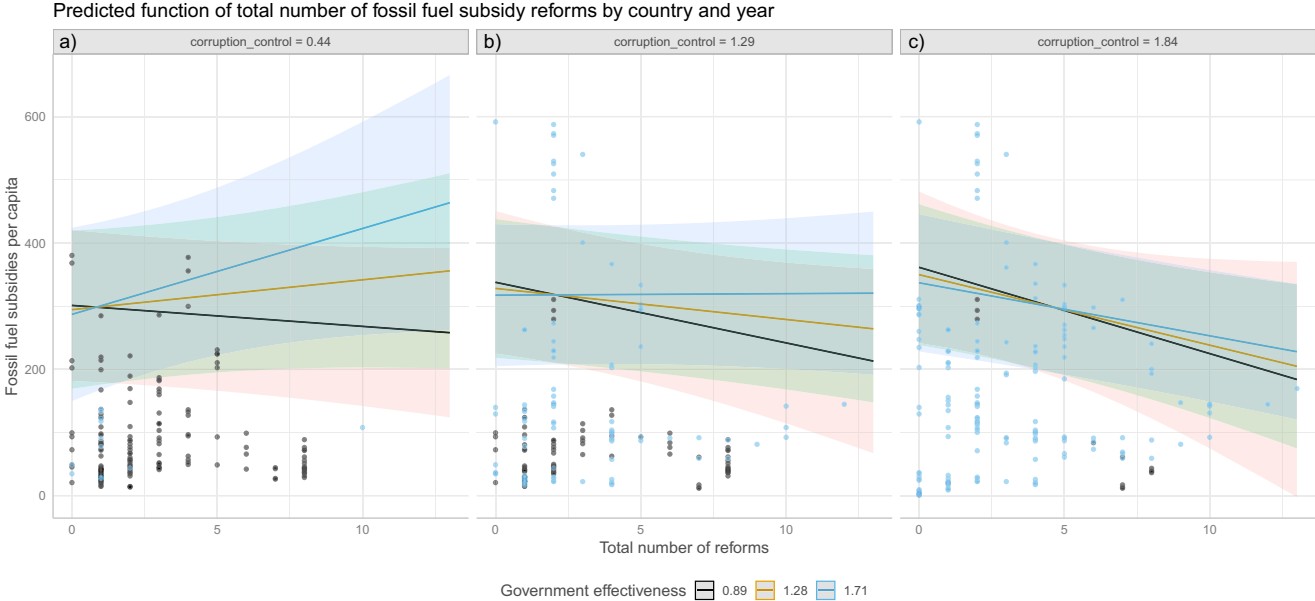

**Fig. 6 | Effect heterogeneity in the polity mechanism.** The relation of total subsidies per capita with number of reforms is conditional on government effectiveness and corruption control. To indicate the interaction effects between corruption control and government effectiveness, the plot shows the estimated functional relation at three different values of corruption control (25%, 50%, and 75% percentiles, among facets from left to right) and for three different values of government effectiveness (with black indicating a value of 25% percentile, yellow indicating the median, and blue indicating 75% percentiles). Raw observations are spread based on below median, between 25% and 75% percentile and above median corruption control among facets **a**–**c** and colored by black for below median government effectiveness values and blue for above median government effectiveness values. Confidence intervals are shaded areas.

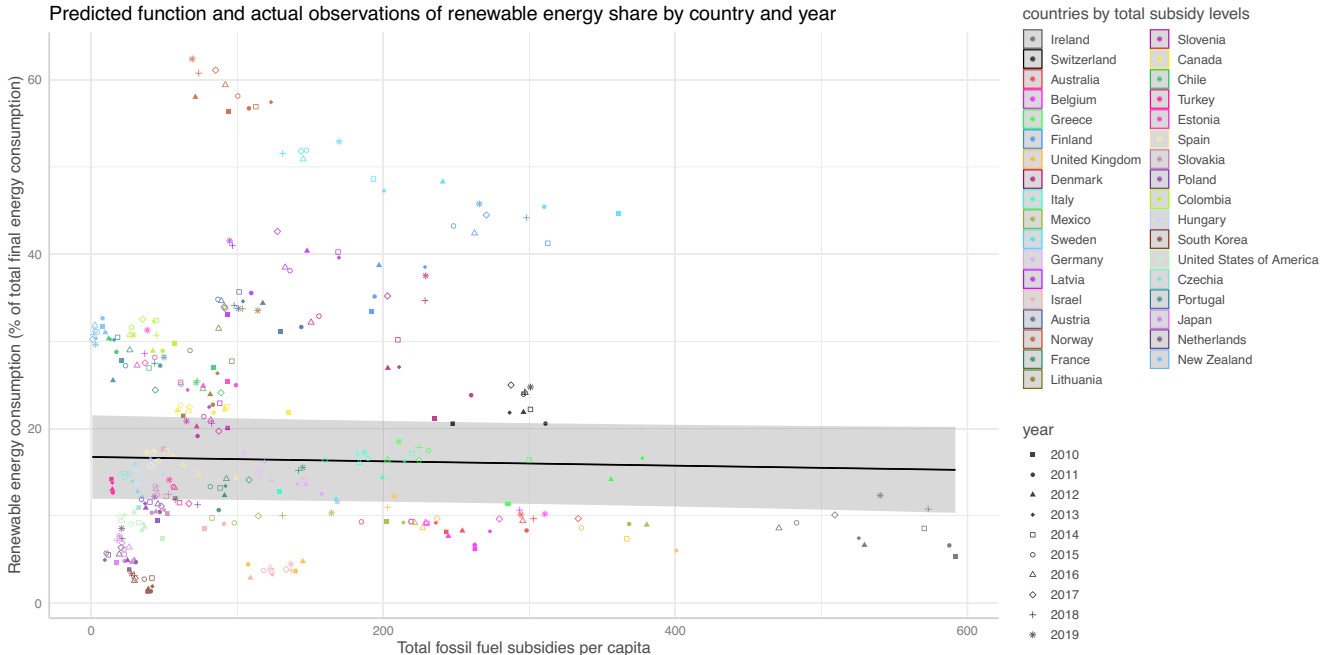

**Fig. 7 | Evidence of feedbacks and entrenching lock-ins.** The figures shows the relation of total subsidies per capita and the share of renewable energy consumption. In addition to the estimated functional relation, the plot shows country and year-based observations. Confidence intervals are shaded areas.

factors influencing fossil fuel subsidies in developing countries. Most notably, our analysis found that the political institutions of democracy and corruption matter for fossil fuel subsidies in industrialized countries, as existing research have already shown that they do in developing countries[24,47]. While the findings are not directly comparable, since the two groups of countries differ not only in their level of industrial development but also in nature of the fossil fuel subsidies provided, they point particularly to the importance of reducing corruption for removing fossil fuel subsidies in both developed and developing countries. While we thus still need to better understand the exact interaction of multiple institutional features such as democratic responsiveness, government effectiveness and corruption control, we provide evidence that institutional quality generally matters. In addition, we find evidence for a positively reinforcing feedback loop where fossil fuel subsidy reforms reduce the level of subsidies. This in turn increases renewable market shares which then reduces the power of fossil fuel interest groups and makes reforms more likely.

Crucial policy implications result from this research: increasing renewable energy production can not only drive out fossil fuels from the energy market, but also from the political system. With increasing support for renewable energy in the system, political reforms that decrease the support for fossil fuel become more likely. Government effectiveness can influence the outcomes of reforms positively but only if sufficiently high corruption control is in place. In order to dismantle fossil fuel subsidies countries must thus work towards improving institutional quality, in particular the control of corruption and isolation of the regulatory political process from private interests.

## Methods

Our analytical approach is abductive. That means that we started by inductively looking into the empirical patterns: To explore and assess the relationship between both i) the number of fossil fuel subsidy reforms and ii) the amount of fossil fuel subsidies, we make use of bagged regression trees[71]. This technique has the advantage to identify highly explanatory variables while allowing for non-linearities and heterogeneous effects in different sub-samples of the population. As a second step, we develop a theoretical model from these empirical insights about important variables and a review of the literature (see

Introduction). The derived hypothesis we then deductively test with the data. Together this approach can be called abductive as it has both indicative and deductive elements. Here, we provide information on the data, and the analytical method including inductive and deductive model specifications. Data and code will be open access.

### Fossil fuel subsidy reforms

The dataset on fossil fuel subsidy reforms covers 35 OECD countries. Costa Rica, Iceland and Luxembourg are OECD countries but have been excluded. Costa Rica, due to its membership being confirmed after the study had begun, and Iceland and Luxembourg due to their populations being below 1 million.

The dataset only includes reforms which remove or reduce support for fossil fuels. This extends to excluding those subsidies that were removed but immediately replaced by a similar subsidy. For example, a tax break on domestic sales being switched to a waiving of export taxes would not be included. In this scenario, while the domestic price would likely rise, government support for fossil fuel production remains unchanged. As no attempt to reduce support for fossil fuels has been made, it is not included in the dataset. Many reforms were also either quickly rolled back or were only partially enacted. To account for this issue, the dataset only includes those reforms that were implemented for at least one full year. Those reforms that were only partially implemented have been entered into the dataset, but a note is made on the failure to fully implement. The description of the reform describes the version of the reform that was implemented. These choices were made to ensure the study would be capable of investigating those conditions under which fossil fuel subsidy removal or reduction occurs. The dataset is not capable of measuring the overall level or number of subsidies in the system (we obtained the data about the levels of subsidies from a different data set as outlined below). Countries may add separate, new subsidies or increase support for existing ones without this being captured within the dataset. This limits how the dataset can be utilized, it can only help to explain when removal or reduction occur – not chart the full removal of national level fossil fuel subsidies. Within this paper we use the dataset to first measure how levels of renewable energy consumption impacted the number of reforms. Second, the impact of

reforms on the subsidies per capita in the system, and third, how these impacts interact with government effectiveness and corruption control. In each of these tests we only observe trends in relation to the number of reforms carried out, as such our observations only describe the reform process and its impacts under different conditions. It is important to note, that as we only record reforms that reduce or remove subsidies, increases to subsidies per capita with increased numbers of reforms (as seen in Fig. 6), may be the result of additional subsidies added to the system or increases in the values of other, separate subsidies. We argue this indicates capture of the reform process by fossil fuel interest groups.

The dataset was built from a variety of sources, for each country a set of core sources was consulted where these core sources were not available, they were skipped. Following this more exploratory data gathering took place utilizing government reports, budgets, academic articles and books, legislative libraries and work from journalists though this would always be then clarified by a second source. First, the OECD maintains a database on fossil fuel subsidies which was consulted for each country. This often contained descriptions of changes to subsidies and the dates these changes took place. Second, the Europan Union (EU) maintains a website with the excise tax rates for fuels and includes any exceptions or lower rates offered to certain sectors or groups. For those non-EU countries in the sample alternatives are sought, usually in the national legislative bodies library of passed laws. Third, national budgets, where available are checked. Fourth, any reports on fossil fuel subsidies produced by the country or through a peer review process are located and checked. Following this we undertake more exploratory research on a country, this involves identifying the structure of the energy market (who owns distribution, transmission etc.), identifying government ministries in charge of industry or any fossil fuel extraction, searching for academic case studies upon the country in questions fossil fuel industry or subsidies and finding any major reforms or collection of reforms that have attracted public attention through journalistic work. Through each of these steps we identified relevant legislation, government owned entities and budgetary practices through which subsides are distributed and build a history of their reform. While this process attempts to be all encompassing there are potential weaknesses. Countries with many well defined, targeted subsidies will likely see more reforms recorded than those that utilize large blanket consumption subsidies simply by having more separate subsidies to dismantle. We attempt to account for this issue by the inclusion criteria outlined below.

When considering different subsidies, we first identify which fuel or energy source is targeted. In general, these subsidies are specific to a certain type of fuel e.g., reduced tax rates on diesel for agricultural use. Electricity where it is directly subsidized, for example consumption subsidies for households or reduced taxation for high energy industry, have also been considered as a potential fossil fuel subsidy. Generation is however, often achieved at least partly through renewables. Reforms to these subsidies have only been considered *fossil fuel* subsidy reforms when generation was primarily achieved by burning fossil fuels. We utilize a cut-off point of at least 60% of generation coming from fossil fuels for electricity subsidies to be considered fossil fuel subsidies. This percentage was set as it ensured that we could identify reforms to subsidies that primarily and consistently benefitted fossil fuels. Setting at the 50% line was avoided as fluctuations in generation year on year would mean subsides often benefited renewables more. Those subsidies that seek to subsidize certain types of electricity generation (hydropower, natural gas etc.) are not subject to this 60% constraint as they are considered a subsidy for the utilized fuel, not the electricity.

The dataset also only considers national level reforms. Those subsidies that are provided at a sub-national level, for example those provided by singular states within federated Canada, have not been

considered within the bounds of the study. This decision has been made as countries have, and continue to be, the main actors within international climate governance, subject of commitments to reform fossil fuel subsidies (e.g. within the G20, the Sustainable Development Goals and the United Nations Framework Convention on Climate Change) and providers of fossil fuel subsidies. While some important initiatives are taken at lower or higher political levels (European Union, United States of America Mayors etc.), international climate regimes remain primarily attached to country level commitments. The focus on the national level also leads the dataset to exclude any reforms of *international* fossil fuel subsidies. By international fossil fuel subsidies we refer to any fossil fuel subsidies that are provided for a third country. Whether through foreign aid, investment or the direct provision of fossil fuels to a third country at no or cut costs (e.g. Venezuela's provision of petroleum products to Colombia at half world prices[72]) these subsidies have not been considered.

As the database intends track those reforms that can be considered a major step towards the dismantlement of fossil fuel subsidies some constraints were applied. These constraints are intended to avoid the inclusion of minor changes in yearly budgets or changes not reflecting a political choice to reform. For example, if a public transport fuel subsidy is not fully utilized one year the budget for the next may see a slight downward realignment towards actual expected cost. Similarly, a government may choose to slightly lower the subsidy on liquefied natural gas fuel for cars but as they have very low uptake this has little impact on subsidy size. We do not consider these serious attempts at reform and as such set certain constraints to avoid their inclusion. Minor changes to very large subsidies however can often be contentious and require considerable political capital to carry out. A 5% drop in agricultural fuel subsidies for example can result in large drops to the countries fossil fuel subsidies per capita. As such we wish to include these reforms despite the relatively small percentage drop in the value of that subsidy. The constraints we have then dictate that subsidies with smaller relative importance, measured by their percentage of the national budget, must have a larger percentage reduction in their overall value. These rules for inclusion are as follows: If the value of a subsidy is at or above 1% of the national budget the reform must represent a 5% drop in value of that subsidy. If the value of a subsidy is between 0.05% and 1% of the budget the reform must represent a 20% drop in value of that subsidy. If the value of a subsidy is below 0.05% of the national budget the reform must represent a 50% drop in value of that subsidy. The complete dismantling of a subsidy is always included regardless of value. If a subsidy changes form in a 'positive' manner, for example a tax exemption changing to a tax reduction, it is always included regardless of value.

Often countries will subsidize fossil fuels by setting their prices at a fixed level for end consumers with the government paying the difference between the fixed price and the global market price. We refer to these subsidies as "fixed fuel prices". Due to the volatile nature of oil and gas prices as well as the different relevance of price indexes to different regions the value of these subsidies is constantly changing, with governments contributing more when prices are high and less when they are low. As such the criteria we outlined above is not suitable for identifying reforms of these subsidies as subsidy values are always highly variable. Instead, if fixed fuel prices increase by more than 10% during a given year, i.e. the government has raised the price for consumers, this has been regarded as a significant reform and has been included. This simple rule is complicated by the presence of price fixing formulas that have been devised in a multitude of iterations across our sample. Price changes as a result of an automatic price adjustment formula have not been included as their 'automatic' nature ensures there is no political will behind the change. Changes to the formula itself however have been included as reforms as long as these alterations are designed to reduce the overall burden of fossil fuel subsidization. Automatic pricing mechanisms have been prone to

discretionary pricing and political interference however. In cases where there is evidence of the mechanism not being applied as intended within a year of adoption the reform has not been included as it is understood to have been rolled-back/failed.

The database has also considered reforms that have not required new or modified legislation, such as the removal of subsidies from a yearly budget. In these cases however reforms were only entered if they had been repeatedly renewed in the past. 'Repeatedly renewed' is considered to be any initiative that has been in place for at least five years and has been renewed at least once. For example, a tax break applied to natural gas that has been renewed yearly for 5 years before being allowed to expire would be included; one that was introduced for a five-year period and then allowed to expire would not be. Again, this rule has been introduced in the interest of identifying political will for subsidy reform. It has been relatively common for subsidies to be introduced with a set expiry date. The aim is often to encourage certain sectoral and regional development or to deal with price shocks and periods of crisis. As our theory concerns structural support for fossil fuels that becomes 'locked-in' it would not be appropriate to include finite extraordinary subsidies that may skew our results. Many of these subsidies however become de facto permanent, as renewal becomes a matter of course rather than question. We attempt to capture these subsidies through the renewal rule. In short, the rule as devised attempts to identify those subsidies which were potentially politically to let expire as opposed to those that simply expired as expected. This rule however produces some limitations in the dataset. Countries that favor supporting fossil fuels through highly targeted but time limited support or through financing specific projects rather than continuous assistance to a sector will be underrepresented in the dataset even if support is reduced over time. Yet, Japan was the only country within the dataset where short term or project-based support was common at the national level. Within this paper however, we seek to understand how more permanent subsidization is phased out and reformed rather than support that ends as projects are completed.

If a reform passes the criteria outlined above it is then placed into one of the following categories: "Instrument change": the means by which a subsidy is implemented has changed to one which aims to reduce the overall subsidy. For example, changing a set fuel price to an automatic pricing mechanism. "Price increase": an increase to a set fuel price. "Change in recipient target group": the subsidy recipients have changed to a more targeted group. "Reduction in reduced tax rate": a reduced tax rate on a fuel for a certain use or industry is brought closer to the regular tax rate for that fuel without achieving parity. "Full dismantling of subsidy": the complete removal of the subsidy. "Tax exemption to reduced tax rate": a tax exemption on a fuel for a certain use or industry is removed but is replaced by a reduced tax rate. "Budgetary transfer decrease": a country making regular budgetary transfers to compensate for shortfalls in cost recovery on fossil fuels reduces the ordinary transfer. For example, a country that requires gas costs to be held low in isolated regions may transfer from the budget to finance the provider's shortfall. If this transfer is reduced, it is included here. "Change in quotas for receiving subsidized fuel": a country lowers the amount of subsidized fuel a recipient is entitled to. "Other": this category captures any remaining reforms that cannot be easily included elsewhere. These reforms tend to be novel as the underlying subsidies are often uncommon methods of subsidizing fossil fuels.

Figure 8 shows how the different types of fossil fuel subsidy reforms are distributed across countries and Table 1 shows how the reforms are distributed in time. For visibility purposes we have collapsed the more fine-grained instrument changes into a single "instrument change category" in Fig. 8. The accompanying data set maintains the higher resolution.

The amounts or levels of fossil fuel subsidies for each country was obtained from the OECD and IISD's online data base "Fossil Fuel Subsidy Tracker"[1]. Their database is a compiled from three complementary international sources and can be considered conservative estimates, if anything there is an under-estimation of their amount (ibid.).

### Set of explanatory variables
Socio-economic data is downloaded from the World Bank's World Development Indicator Database[73]. In particular, we use gross domestic product "GDP per capita growth (annual %)", "$CO_2$ intensity (kg per kg of oil equivalent energy use)", "Renewable energy consumption (% of total final energy consumption)", and "Fuel exports (% of merchandise exports)". Crude oil prices are downloaded from Our World in Data[74]. The domestic production of fossil fuels is gathered from BP's Statistical Review of World Energy, in particular = "Oil Production – Barrels", "Gas Production – Bcm", and "Coal Production – Tonnes"[75]. State owned oil, gas and coal production was originally drawn from Cheon, Lackner & Urpelainen's data on national oil companies[21]. This data was then expanded with the addition of the state-owned coal company dummy as well as including state owned production companies for Poland and Austria. Green party family votes shares per country and election year are gathered from the ParlGov database[76] and vote shares are held constant until the next observed election. Information on the "quality of institutions" concept was operationalized with data collected from the Quality of Government database[77]. Here, we use the indicators "Electoral democracy index", "Public sector corruption index", "Liberal component index", "Legislature corrupt activities", "Executive corruption index". Additionally, we also use information on the government effectiveness from the Worldwide Governance Indicators[78] that includes perceptions of the quality of public and civil services and the degree of independence from political pressures. Finally, we use a dummy variable to indicate the Paris Agreement and government surplus is measure in % of GDP, 2000–2021, based on OECD data.

### Abductive analysis
The overall empirical analysis consists of an inductive, exploratory analysis to understand patterns and variable importance based on observations and deductive theory testing. Through iterations, the resulting analysis can be considered abductive in its approach.

The inductive, exploratory analysis is conducted in three steps to inform theory building: through a regression tree, multiple visual inspections with a random variation, and finally a tree bagging approach to derive variable importance (see Supplementary Figs. 1–5). For all these steps we use three different outcome variables, that is either the number of fossil fuel subsidy reforms, the amount of fossil fuel subsidies, or the amount of production based $CO_2$ emission (see data above). We let this be denoted by $Y_{i,t}$ as the outcome variable in country $i$ and year $t$. The outcome is predicted as a constant mean outcome $c_b$ in region $R_b$ ($\bar{y}\forall i \in R_b$) for possible combinations of explanatory variables[71]. $X_{i,t}$ is a vector of explanatory variables, including the remaining variables of interest that function as outcome variables in other sets of regressions, e.g. number of subsidy reforms and $CO_2$ emissions in case the dependent variable is the amount of fossil fuel subsidies and vice versa. Furthermore, $X_{i,t}$ includes the controls specified above, and it additionally covers a running year variable to account for general change over time. For the regression trees, we use the following specification for the rpart package[79], where a single tree is a function of the explanatory variables $X_i$:

$$f(X_{i,t}) = T(X_{i,t}; \Theta) \equiv \sum_{b=1}^{B} c_b + I(X_{i,t} \in R_b) \tag{1}$$

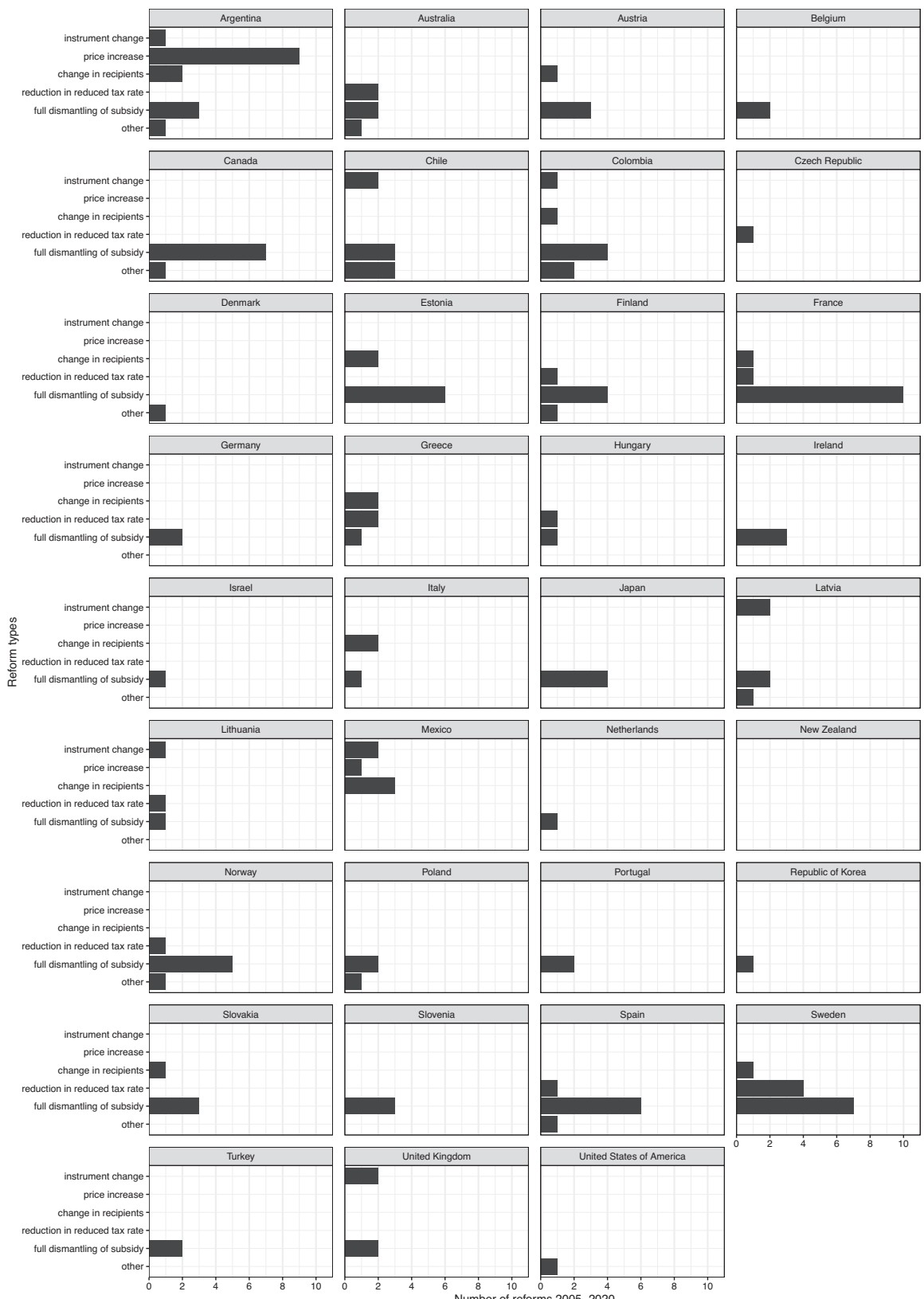

**Fig. 8 | Quantity and types of fossil fuel subsidy reforms by country within the sample.** The numbers indicate the occurrences of different fossil fuel subsidies over a 2005–2020 period. The goal of all counted reforms is to reduce fossil fuel subsidies. Categories include 'instrument changes', 'price increases', 'change in recipients', 'reduction in reduced tax rate', and 'full dismantling of subsidy', plus a miscellaneous 'other'. For further details on the typology see Methods.

**Table 1 | Summary statistics of the collected fossil fuel subsidy reform database. Values indicate the number of reforms per year**

| | 2005 | 2006 | 2007 | 2008 | 2009 | 2010 | 2011 | 2012 | 2013 | 2014 | 2015 | 2016 | 2017 | 2018 | 2019 |
|---|---|---|---|---|---|---|---|---|---|---|---|---|---|---|---|
| Price increase | 0 | 0 | 0 | 0 | 0 | 0 | 0 | 0 | 0 | 0 | 0 | 1 | 0 | 0 | 0 |
| Instrument change | 0 | 0 | 0 | 1 | 0 | 1 | 1 | 0 | 0 | 1 | 0 | 0 | 1 | 0 | 0 |
| Full dismantling of subsidy | 4 | 3 | 7 | 4 | 6 | 11 | 10 | 10 | 9 | 5 | 7 | 3 | 8 | 1 | 5 |
| Change in recipient target group | 0 | 0 | 2 | 0 | 0 | 2 | 2 | 1 | 0 | 1 | 2 | 1 | 0 | 0 | 0 |
| Reduction in reduced tax rate | 0 | 0 | 1 | 1 | 5 | 1 | 6 | 3 | 0 | 0 | 0 | 1 | 0 | 0 | 2 |
| Tax exemption to reduced tax rate | 0 | 0 | 0 | 0 | 0 | 1 | 2 | 0 | 0 | 0 | 2 | 0 | 0 | 0 | 0 |
| Budgetary transfer decrease | 0 | 0 | 0 | 0 | 0 | 0 | 0 | 0 | 0 | 0 | 0 | 0 | 0 | 0 | 0 |
| Change in quotas for subsidized fuels | 0 | 0 | 0 | 0 | 0 | 0 | 0 | 0 | 0 | 0 | 0 | 0 | 0 | 0 | 0 |
| Other | 1 | 1 | 0 | 0 | 0 | 0 | 2 | 1 | 2 | 1 | 0 | 0 | 1 | 1 | 1 |
| Total | 5 | 4 | 10 | 6 | 11 | 16 | 23 | 15 | 11 | 8 | 11 | 6 | 10 | 2 | 8 |

**Table 2 | Regression results**

| | Model 1 | Model 2 | Model 3 | Model 4 | Model 5 |
|---|---|---|---|---|---|
| Dependent variable | Number of reforms | Fossil fuel subsidies per capita | Number of reforms | Fossil fuel subsidies per capita | Renewable energy shares |
| (Intercept) | 10.29*** (1.74) | 607.53*** (128.00) | 5.40*** (1.41) | 834.09*** (153.25) | 32.27*** (5.47) |
| renewable_share | 0.08*** (0.01) | −0.59 (0.74) | −0.36*** (0.09) | | |
| total_subsidies_cap | −0.00* (0.00) | | | | −0.00* (0.00) |
| gdp_cap | −0.00*** (0.00) | −0.00 (0.00) | −0.00 *** (0.00) | −0.01 ** (0.00) | 0.00 (0.00) |
| I(gdp_cap^2) | 0.00*** (0.00) | 0.00 (0.00) | 0.00 *** (0.00) | 0.00 *** (0.00) | 0.00 (0.00) |
| ghg_prod_emiss_cap | 0.18*** (0.06) | −0.83 (4.50) | 0.38 *** (0.05) | 1.40 (3.28) | −1.33*** (0.12) |
| democracy_qual | 3.80*** (0.53) | −64.26 (52.41) | −0.74 (0.99) | −30.31 (59.38) | 6.14 (4.05) |
| total_number_reforms | | −7.05 *** (2.04) | | −22.39 *** (3.55) | |
| crude_oil_price | | −1.11 (0.69) | | −1.95 ** (0.87) | −0.16*** (0.02) |
| public_corruption | | −122.72 (146.22) | | −82.53 (172.58) | 0.16 (6.67) |
| legislative_corruption | | 26.52 (19.02) | 0.97 *** (0.28) | 17.57 (16.90) | −0.31 (0.72) |
| executive_corruption | | −105.87 (73.52) | | −94.14 (68.59) | −1.75 (2.61) |
| liberal_dem | | −179.39 * (91.26) | | −213.74 ** (103.21) | −1.27 (4.57) |
| government_effectiveness | | 31.39 ** (14.20) | | −13.32 (29.07) | 0.40 (0.71) |
| corruption_control | | | | 51.05 (36.20) | 2.27*** (0.67) |
| renewable_share: democracy_qual | | | 0.52 *** (0.12) | | |
| total_number_reforms: corruption_control | | | | 1.64 | |
| | | | | (6.85) | |
| government_effectiveness: corruption_control | | | | −8.97 (19.53) | |
| total_number_reforms: government_effectiveness | | | | 25.08*** | |
| | | | | (7.18) | |
| total_number_reforms: government_effectiveness: corruption_control | | | | −10.14*** | |
| | | | | (2.73) | |
| Country fixed effects | Yes | Yes | Yes | Yes | Yes |
| Year fixed effects | Yes | Yes | Yes | Yes | Yes |
| Nobs | 350 ($n = 35$, $t = 10$) | 350 ($n = 35$, $t = 10$) | 525 ($n = 35$, $t = 15$) | 350 ($n = 35$, $t = 10$) | 350 ($n = 35$, $t = 10$) |
| adj.r.squared | 0.89 | 0.91 | 0.79 | 0.91 | 0.99 |
| F-statistic | 59.70 | 64.59 | 37.49 | 63.67 | 463.51 |
| *p*-value | 0.00 | 0.00 | 0.00 | 0.00 | 0.00 |
| df | 49.00 | 54.00 | 55.00 | 58.00 | 54.00 |

***$p < 0.01$; **$p < 0.05$; *$p < 0.1$.

To optimize the predictive power of a single regression tree we prune the tree by running a grid search over a range of hyperparameters Θ such as minimal number of splits, maximal tree depth, the maximum number of surrogate variables, and a complexity parameter, by optimizing $\hat{\Theta} = \arg\min_{\Theta} \sum_{b=1}^{B} \sum_{X_{i,t} \in R_b} L(y_i, c_b)$[71] with loss function $L$ that minimizes the cross-validation error computed through the build in cross-validation function from rpart, calculating the predicted residual error sum of squares[79]. To increase the robustness and validity of the analysis we compute n-fold cross-validated decision with trees voting in an ensemble, using the *caret* package[80]. This reduces the

variance in single-tree estimates of the multi-dimensional response or outcomes surface by fitting multiple trees and combining them based on bootstrap aggregation. Namely, random subsamples of the original data are drawn with replacement through bootstrapping. A non-pruned tree is fitted to each sample and the bootstrap model is $\hat{f}_{bag}(X_{i,t}) = \frac{1}{M}\sum_{m=1}^{M} T(X_{i,t}; \hat{\Theta}_m)$ for $M = 1000$ trees. To address the longitudinal structure of the data and the possibility of leakage in the cross-validation process, we employ a grouping, such that the number of folds equal the number of countries per bagged tree regression and the repeated measurements per country are indexed accordingly. While bagging has the disadvantage of correlated trees compared to random forests, the corresponding R functions within *rpart* allow for using surrogates at splits in case single values are unobserved which in turn minimizes data loss. Results from the regression trees for inductive pattern analysis can be found in the Appendix.

For the deductive part, we run two-way fixed effects models that account for unobserved heterogeneity among countries, and shared temporal dynamics such as cultural shifts and unaccounted for technological advances. The two-way fixed effects model can be considered a fairly robust approach to causal inference on observational data[81] but its validity hinges on model specifications[82]. We therefore only proceeded with parametric estimations once we had concluded the inductive phase of hypotheses generation. However, the theoretical model building continued during this phase and the presented theory was further refined even if basic patterns had already been established from data-driven pattern analysis through bagged regression trees. While the main results are reported as figures in the main text, here we provide further details on the quantitative causal inference strategy. In general, two-way fixed effects regressions take the following form[81]

$$y_{it} = \alpha_i + \beta x_{it} + \lambda_t + u_{it}, t = 1, \ldots, T \ i = 1, \ldots, N, \quad (2)$$

where $\alpha_i$ are unit specific intercepts and $\lambda_i$ are time specific effects. Our independent variables of interests $x_{it}$ are the explanatory variables described above, and we estimate their effect on different outcome variables $y_{it}$ to test our derived hypothesis (see Fig. 2 in the main text for an overview of hypotheses). Due to data availability constraints in the covariates our final dataset comprises a balanced panel data set of $N = 35$ countries and $T = 10$ years (except for model three were we have 5 more years of data because fossil fuel subsidy levels is not included). We employ a stepwise variable selection to estimate sparse models based on an Akaike-Information Criterion. In all models, we include unit-specific time averages and time-period specific cross-sectional averages or so-called country-fixed and year-fixed effects in the form of corresponding dummy variables. This allows us to estimate within country variations in the outcomes. For standard errors we compute clustered covariance estimations for panel data based on procedures developed by Newey-West[83] and Driscoll and Kraay[84] as implemented in the R package sandwich[85]. These are robust to heteroskedasticity and autocorrelation and are clustered at country and year level. In Table 2, we report the regresion results. Model 1 corresponds to Fig. 3 in the article, Model 2 to Fig. 4, etc. For the moderating effects of institutional quality and the feedback mechanism we estimated two and three way interaction models (see Table 2, Model 3 and 4 for details)[65]. To support our argument for effect heterogeneity imposed by institutional variation, we also provide a table of partial derivatives for total numbers of reforms, computed across a grid of quantiles of both government effectiveness and corruption control, across all countries, holding the other variables constant (Supplementary Data 1).

## Data availability

The collected compiled data can be found here: https://doi.org/10.5281/zenodo.10320470.

## Code availability

Code for generating the cleaned dataset, and the analytical code for replicating results and figures in this article can be found here: https://doi.org/10.5281/zenodo.10320470.

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

## Acknowledgements
We would like to thank Sverker Jagers, Niklas Harring, Nina von Huexkull, the participants at the Centre for Collective Action Research meeting in Gothenburg, and Participants at the International Studies Association 2022 meeting. B.C. and J.S. have received funding from the Swedish Energy Agency grant number P48761-1. This paper is a contribution to the Strategic Research Area "Biodiversity and Ecosystem Services in a Changing Climate" (BECC), funded by the Swedish government.

## Author contributions
N.D. data cleaning, inductive analysis, analytical code and formal analysis, figure design, theory development, writing of draft, revisions, and final version; B.C. data curation, data analysis and interpretation, theory development, writing of draft, revisions and final version; J.S. conceived the original research idea, data interpretation, theory development, writing of draft, revisions and final version, provided funding.

## Funding

## Competing interests
The authors declare no competing interests.
