## [Peer Review File · Nature Communications]

REVIEWER COMMENTS

Reviewer #1 (Remarks to the Author):

In this academic article, the authors address the persistent challenge posed by fossil fuel subsidies in the context of meeting the goals of the Paris Agreement. They emphasize the critical need to comprehend the political and economic aspects of these subsidies and their potential reform. To explore these dynamics in developed countries, the authors employ a newly created database containing various types of fossil fuel subsidy reforms implemented in OECD countries. They aim to develop and validate a theory concerning governance mechanisms. The dataset includes and categorizes fossil fuels subsidy reforms by type and covers subsidy initiatives ranging from tax expenditures and direct budgetary transfers to royalty exemptions and price fixing. The authors argue that they make two key contributions: (1) and empirical by providing new data and (2) a theoretical one by proposing which mechanisms could lead to reforms of fossil fuel subsidies.

The presented dataset categorizes fossil fuel subsidy reforms by type, encompassing initiatives ranging from tax expenditures and direct budgetary transfers to royalty exemptions and price controls. The authors contend that their contributions manifest in two key dimensions: firstly, the provision of novel empirical data, and secondly, the presentation of a theoretical framework elucidating the mechanisms that could stimulate reforms pertaining to fossil fuel subsidies.

While acknowledging the utility of the data collection effort and generally commending the article's prose, I am compelled to voice a number of substantial reservations. My foremost concern revolves around the overarching research design. Given the relatively limited dataset, which spans 35 countries over a decade, I have doubts regarding the appropriateness of the chosen methods, which commence with an inductive approach using random forest analysis followed by fixed effects regression. It appears that these methodologies might not yield sufficiently robust insights. To enhance the article's utility, I suggest augmenting the descriptive evidence. For instance, Figure 1 could be supplanted with bar plots that illustrate, for each country, both the quantity and the type of reforms. Additional more descriptive avenue could be a focus on country clusters based on variables such as government effectiveness, corruption levels, and the prevalence of incumbent industries (renewable versus fossil). Such an approach could serve to elucidate variations in reform levels and could be supported by in-depth case studies.

The article's focus on OECD countries, where government effectiveness exhibits limited variation, raises concerns that the findings may be skewed by a few outliers. In this context, I am not sure that the role of government effectiveness should be highlighted so much (or maybe could be if the authors add other countries to the sample).

Regarding the theoretical component, while it is well-conceived, there is an opportunity for the authors to engage with a broader literature on the political economy of energy. Numerous similar frameworks have been established in previous research (e.g., Aklin & Urpelainen, 2013; Cherp et al., 2018), which could enrich the current discussion. Given the limited dataset, I would recommend de-emphasizing the inductive approach, such as the utilization of random forest analysis to identify key variables, and instead advocate for grounding the theoretical framework in existing literature to bolster its legitimacy.

Aklin, M., & Urpelainen, J. (2013). Political Competition, Path Dependence, and the Strategy of Sustainable Energy Transitions: SUSTAINABLE ENERGY TRANSITIONS. *American Journal of Political Science*, 57(3), 643–658. <https://doi.org/10.1111/ajps.12002>

Cherp, A., Vinichenko, V., Jewell, J., Brutschin, E., & Sovacool, B. (2018). Integrating techno-economic, socio-technical and political perspectives on national energy transitions: A meta-theoretical framework. *Energy Research & Social Science*, 37, 175–190. <https://doi.org/10.1016/j.erss.2017.09.015>

Reviewer #2 (Remarks to the Author):

The study entitled "A Political Economy Theory of Fossil Fuel Subsidy Reforms" introduces an article that presents a new dataset mapping out fossil fuel subsidy reforms across 35 OECD countries from 2005-2020. The authors claim this dataset is used to abductively develop a theory on the political economy of fossil fuel subsidy reforms. However, several significant shortcomings are found regarding the claim and presentation of the manuscript. Novelty, methodology, and the modeling process need special attention. The robustness of the modeling and results still has substantial problems (or weaknesses) that need to be explained clearly. From all of these shortcomings, this manuscript has doubtful quality for publication in *Nature Communication* unless significant changes are made. The authors are advised to address the following comments carefully (please find the detailed comments attached).

Reviewer #3 (Remarks to the Author):

The authors propose an analysis of fossil fuel subsidy reforms developed under the lens of political economy theory. They employ an abductive approach to identify the relevant variables and develop a theory about causal mechanisms based on existing literature. Then the hypotheses derived from

the theory are empirically tested on a new dataset, mapping fossil fuel subsidy reforms from 35 OECD countries from 2005-2020.

More specifically, they investigate three mechanisms:

- i) a market-power mechanism, according to which larger renewable energy shares ease the reform of fossil fuel subsidies (RH1: larger renewable energy shares, higher number of reforms);
- ii) a policy mechanism, according to which reforms lead to lower overall fossil fuel subsidy levels (RH2: higher number of reforms, lower level of subsidies);
- iii) a feedback mechanism, according to which mechanisms i) and ii) are affected by institutional quality (i.e., positively influencing both the positive correlation in RH1 (RH3.1), and the negative correlation in RH1 (RH4)) and path dependence (i.e., negatively influencing the positive correlation in RH1 (RH3.2)).

The model is estimated by means of panel fixed effect model. A preliminary inductive exploratory analysis is run through a regression tree, multiple visual inspections with a random variation, and finally a tree bagging approach to derive variable importance.

My first concern is related to how the data selection and the construction of the database may drive the results. This is because the dataset on the number of reforms only includes those interventions that remove or reduce support for fossil fuels, conditional on additional inclusion constraints (e.g., “If the value of a subsidy is at or above 1% of the national budget the reform must represent a 5% change in value of that subsidy”) as detailed at pag. 1 in the Supp. Material. In this respect, the authors should make a larger effort to explicitly discuss their selection criteria, the motivation behind and possible bias determined by these choices (particularly with respect to the hypothesis tested and endogeneity concerns in the empirical analysis). In the current form, the paper seems more about fossil fuel subsidy removal or reduction, rather than the more general idea of fossil fuel subsidy reform. In addition, it is not clear from which sources information have been collected, and I believe this is an important aspect since the authors emphasise the novelty of the dataset.

My second main comment is related to the interpretation of the results, especially with respect to the evidence provided for testing the feedback mechanisms, operationalised through the introduction of different interaction terms in the model specifications. The interpretation of marginal effects is not trivial when introducing different interactions, but the paper does not provide the reader with any guidance. The paper would largely benefit from discussing the estimated coefficient of the interactions (and their statistical significance), indicating why (or why not) results are consistent with theory. More generally, the authors should make a larger effort to discuss their estimations and explain their correspondence with the research questions addressed in the paper. For instance, in model 3 table A2 there are three interaction terms (i.e., `renewable_share:total_subsidies_cap`, `renewable_share:government_effectiveness`, `total_subsidies_cap:government_effectiveness`) and only the third one is statistically significant. The authors suggest that their findings support RH3.2, a conditional support for RH3.1 and mixed evidence for RH4. Unfortunately, the paper does not provide any explanation for these results.

In addition, few minor points:

- No reference list in the supplementary materials

- It is not clear the time span covered: the new dataset (should) cover the period from 2005 to 2020, but different time periods are reported in different figures (e.g., 2000-2020 in figure 1; 2010-2019 in figures 3-4).

Comments on NCOMMS-23-39793-T

The study entitled "A Political Economy Theory of Fossil Fuel Subsidy Reforms" introduces an article that presents a new dataset mapping out fossil fuel subsidy reforms across 35 OECD countries from 2005-2020. The authors claim this dataset is used to abductively develop a theory on the political economy of fossil fuel subsidy reforms. However, several significant shortcomings are found regarding the claim and presentation of the manuscript. Novelty, methodology, and the modeling process need special attention. The robustness of the modeling and results still has substantial problems (or weaknesses) that need to be explained clearly. From all of these shortcomings, this manuscript has doubtful quality for publication in Nature Communication unless significant changes are made. The authors are advised to address the following comments carefully.

A. Formatting

1. The article should be written with clear and structured sections and subsections.
2. The references are written manually; some texts need references but are missing. It is recommended that the authors use referencing software to assist in article writing.
3. There are a total of 69 references in the text (body text and methodology), but in the bibliography, there are only 56 references. The authors should resolve this problem.
4. Most cited academic papers are from the period 2015-2018. It may be more useful if the authors could add more recent references, such as from 2021; the dynamics of the energy sector (renewable energy (RE) vs fossil fuels) have developed more rapidly in recent years.

B. Substance and Body of Analysis

1. If the study focuses on OECD countries, the title should be revised to state it.
2. P2 L77-78: "Policy interventions supporting RE helps to transition the economy away from fossil fuel."

In this citation, policy intervention to RE includes subsidy as well. How could the authors rationalize or differentiate the subsidy for RE and subsidy for fossil fuels in the model?

3. P3 L87-89: "Subsidies provide concentrated benefits for specific interest groups that produce or consume fossil fuels, subsequently increasing their share of the economy."

Fossil fuel subsidy in the Middle East and some Asian countries (such as Indonesia) characterizes universal subsidy, likewise, bioethanol subsidy in developed countries (such as Australia). How could the author differentiate these 2 types of subsidy in the model?

4. Figure 2 P3 describes the structural causal model of how the energy market and political institutions interact when it comes to fossil fuel subsidy reforms. There is a flaw in the concept of this model; the energy market is not the same as (has a larger scope than) the

renewable energy market. Hence, this model needs to be revised to be more representative; what subsystems of the energy market are involved in this model?

5. Figure 2 P3: It is unclear what basis or approach the authors used to build the causal model. This should be explained and is important to be explained in the methodology as well, and how this causal model is combined with the regression and random forest analysis?
6. Figure 2 P3: What do the rectangle and circle symbols mean? If we refer to the system dynamics concept, both symbols represent different things.
7. Figure 2 P3: Based on the concept of causal modeling, a variable will influence another variable not the flow (in the Figure, flow is represented by an arrow sign). Another problem, there is an inconsistency about the arrow from “fossil fuel subsidy levels” **to or from** “lock-in effects.” The authors should explain the rationale for these model-building problems. It cannot be seen clearly how the flow of this figure is translated into the regression model. Meanwhile, each symbol has some proxies, such as the quality of political institutions measured by government effectiveness and corruption level? Figure 2 really needs to be revised.
8. If the causal model as Figure 2 P3 refers to a causal loop diagram (system dynamics), the logic and model building need to be revised.
9. Unclear cross-references or links between the causal model, hypothesis, variables, and measurement, and the regression model. The authors need to provide informative cross-references among these aspects.
10. P4 L125-131: “Our theory consists of three connected mechanisms: i) a market-power mechanism that captures the effects of competition between energy sectors, and where larger renewable energy shares decrease the role of fossil fuels and thereby ease the reform of fossil fuel subsidies; ii) a policy mechanism where reforms lead to lower overall fossil fuel subsidy levels; iii) a feedback mechanism that captures the effects of institutional quality (e.g., government effectiveness, corruption control), lock-in into fossil fuel-dependent pathways, and their subsequent effect on the market-power mechanism as well as the policy mechanism.”

The authors need to confirm clearly what the market-power mechanism, policy mechanism, institutional quality represent? It is unclear if the variables government effectiveness and corruption control are only the examples of institutional quality, so what about many other variables or factors?

11. Hypothesis 1: the larger the market share of alternative technologies, such as renewables, the easier it is to reform fossil fuel subsidies.

Hypothesis 1 has weakness and is not straightforward; there are lots of alternative technology types, meanwhile, in this hypothesis, renewables are only the example? This hypothesis also inconsistent with the causal model built; the renewables are the focus of investigation (not an example).

12. P5 L161-162: The third type of mechanism is placed on the system level and captures how structural aspects of the socio-economic system influence the functioning of the main causal chain. What is the main causal chain in this statement? Is there any supporting causal chain?
13. Hypothesis H1, H2, H3.1, H3.2, and H4 need to be revised/adjusted once comments numbers B9, 10, and C6, 8.
14. H3.2 seems to have an intermediary variable, i.e., the level of fossil fuel subsidy influences the renewables' share via lock-in effects. How the authors could reveal and explain this intermediary calculation?
15. P7 L212-216, the sentences in these lines are unclear, vague, inconsistent, and contradictory between each other; needs to be revised. References are needed as well.
16. The Discussion and Conclusion Section and Abstract need to be revised after all issues highlighted have been resolved. Here, the authors also need to summarize: i) how the methodology contributes to answering the problems being investigated, ii) the answer of hypotheses being tested.

C. Data and Methods

1. Unclear political economy theory approaches used and referred to in this paper (both in body analysis and methods).
2. P1 L1-2: "To explore and assess the relationship between both i) the number of fossil fuel subsidy reforms and ii) the amount of fossil fuel subsidies, we make use of regression trees and random forests." The logic of the random forest cannot be seen in the model or calculation.
3. Sentences on P1 L21-25 seem contradictory and unclear. The authors need to specify what is the proxy or measure that they use and differentiate for electricity subsidy or fuel subsidy? Are both subsidies used in the modeling? What is the judgment of 60% constraint in the subsidy reform?
4. What is the judgment of classifying the constraints of the percentage of subsidy from the national budget as P1 L40-48? The sentence writing is also unclear; needs to be rewritten. Such as: if the value of a subsidy is at or above 1% of the national budget, the reform must represent a 5% change in the value of that subsidy. Still unclear about the relationship and judgment about 1% national budget and 5% change of subsidy value, and the reform itself. Likewise for the subsequent points.

Meanwhile, in P2 L49 stated: These rules (L40-48) cannot be applied to increases in fixed fuel prices. Are the authors able to provide relevant explanations and reasons about this matter? What does "increases in fixed fuel prices" mean? On the other hand, the nature of fuel is fluctuating.
5. How the authors could explain criteria as stated in P2 L73-92 and Table 1 are aligned with the variables in Table 2. What are the relationships between the criteria and variables?

6. What is the unit of measurement in Table 1?
7. The authors need to explain the rationality of selected or chosen variables.
8. The authors need to explain briefly the validation process of modeling and the calculation.

REVIEWER COMMENTS

Reviewer #1 (Remarks to the Author):	Response
In this academic article, the authors address the persistent challenge posed by fossil fuel subsidies in the context of meeting the goals of the Paris Agreement. They emphasize the critical need to comprehend the political and economic aspects of these subsidies and their potential reform. To explore these dynamics in developed countries, the authors employ a newly created database containing various types of fossil fuel subsidy reforms implemented in OECD countries. They aim to develop and validate a theory concerning governance mechanisms. The dataset includes and categorizes fossil fuels subsidy reforms by type and covers subsidy initiatives ranging from tax expenditures and direct budgetary transfers to royalty exemptions and price fixing. The authors argue that they make two key contributions: (1) and empirical by providing new data and (2) a theoretical one by proposing which mechanisms could lead to reforms of fossil fuel subsidies. The presented dataset categorizes fossil fuel subsidy reforms by type, encompassing initiatives ranging from tax expenditures and direct budgetary transfers to royalty exemptions and price controls. The authors contend that their contributions manifest in two key dimensions: firstly, the provision of novel empirical data, and secondly, the presentation of a theoretical framework elucidating the mechanisms that could stimulate reforms pertaining to fossil fuel subsidies. While acknowledging the utility of the data collection effort and generally commending the article's prose, I am compelled to voice a number of substantial reservations.	We thank the reviewer for constructive comments. Please find our detailed responses below
1. My foremost concern revolves around the overarching research design. Given the relatively limited dataset, which spans 35 countries over a decade, I have doubts regarding the appropriateness of the chosen methods, which commence with an inductive approach using random forest analysis followed by fixed effects regression. It appears that these methodologies	We agree that more descriptive information would increase the accessibility of the data and in-depth case studies would indeed be a really good way forward (albeit out of scope for this article). We have therefore provided an additional figure that displays fossil fuel reform types by country in the Methods and Data section. In addition, we point out the need for

REVIEWER COMMENTS

might not yield sufficiently robust insights. To enhance the article's utility, I suggest augmenting the descriptive evidence. For instance, Figure 1 could be supplanted with bar plots that illustrate, for each country, both the quantity and the type of reforms. Additional more descriptive avenue could be a focus on country clusters based on variables such as government effectiveness, corruption levels, and the prevalence of incumbent industries (renewable versus fossil). Such an approach could serve to elucidate variations in reform levels and could be supported by in-depth case studies.	more in-depth studies in the Discussion section and Figure M1 in the Data and Methods points out some particularly interesting candidates for such. Regarding the reliability of results, the random forest model has just been used to provide preliminary indication of variable importance. This has in turn informed our theoretical model structure. The derived hypotheses were then tested through a fixed effects model with clustered, robust standard errors. While there may be some remaining uncertainty regarding asymptotic qualities with a Panel of N=35, T=10, such a panel data size is not uncommon in the literature. One of the most widely used data sets in applied econometrics, the Grunfeld data on firm level investments has only 11 firms and 20 annual observations. Given that fixed effect qualities have been studied extensively with such data set sizes, we would argue that the two-way fixed effects model provides sufficiently consistent estimators. On top we employ clustered standard error á la Discroll and Kraay which are robust to general forms of cross-sectional and serial correlation. We have of course checked for outliers but could not observe any, based on a measure of Cook's distance. The code and data will be made open access to allow for maximum transparency and replicability.
2. The article's focus on OECD countries, where government effectiveness exhibits limited variation, raises concerns that the findings may be skewed by a few outliers. In this context, I am not sure that the role of government effectiveness should be highlighted so much (or maybe could be if the authors add other countries to the sample).	We understand the concern that a few outliers have skewed the results, but in the data set Worldwide Governance Indicators data from The World Bank that that we have used for government effectiveness is a normalized variable between -2.5 and 2.5 (for more info see here: https://www.govindicators.org/). Plotting the distribution of values within our data set shows that we have generally rather effective countries in the data set which is not surprising given that they are OECD countries. We do however observe quite some spread between -0.5 and 2.5 with some higher density towards higher ranges, but all in all not a very skewed distribution. There are no outliers in the data, but

REVIEWER COMMENTS

	of course thinner tails. See the Violin and Box plot below where we also show observations.  The figure is a violin plot with an overlaid box plot. The x-axis is labeled 'government_effectiveness' and ranges from -1 to 2. The violin plot shows a distribution that is roughly bell-shaped but with a slight right skew, peaking around 1.5. The box plot shows the median is approximately 1.2, with the interquartile range (IQR) from about 1.0 to 1.4. Whiskers extend from approximately 0.5 to 2.0. Numerous individual data points are overlaid on the violin plot, showing a concentration between 0.5 and 1.5.
1. Regarding the theoretical component, while it is well-conceived, there is an opportunity for the authors to engage with a broader literature on the political economy of energy. Numerous similar frameworks have been established in previous research (e.g., Aklin & Urpelainen, 2013; Cherp et al., 2018), which could enrich the current discussion. Given the limited dataset, I would recommend de-emphasizing the inductive approach, such as the utilization of random forest analysis to identify key variables, and instead advocate for grounding the theoretical framework in existing literature to bolster its legitimacy. Aklin, M., & Urpelainen, J. (2013). Political Competition, Path Dependence, and the Strategy of Sustainable Energy Transitions: SUSTAINABLE ENERGY TRANSITIONS. American Journal of Political Science, 57(3), 643–658. https://doi.org/10.1111/ajps.12002 Cherp, A., Vinichenko, V., Jewell, J., Brutschin, E., & Sovacool, B. (2018). Integrating techno-economic, socio-technical and political perspectives on national energy transitions: A meta-theoretical framework. Energy Research & Social Science, 37, 175–190. https://doi.org/10.1016/j.erss.2017.09.015	Thank you for this very useful suggestion. While we want to keep the description of the inductive approach, since this was indeed how we proceeded, we have grounded our framework more explicitly in the existing literature. This has been done by expanding on the initial description of the theoretical framework (on p. 4) as well as on the description of the individual hypotheses. The hypotheses are now more clearly related to existing literature, including the two references suggested by the reviewer.

REVIEWER COMMENTS

Reviewer #2	
The study entitled "A Political Economy Theory of Fossil Fuel Subsidy Reforms" introduces an article that presents a new dataset mapping out fossil fuel subsidy reforms across 35 OECD countries from 2005-2020. The authors claim this dataset is used to abductively develop a theory on the political economy of fossil fuel subsidy reforms. However, several significant shortcomings are found regarding the claim and presentation of the manuscript. Novelty, methodology, and the modeling process need special attention. The robustness of the modeling and results still has substantial problems (or weaknesses) that need to be explained clearly. From all of these shortcomings, this manuscript has doubtful quality for publication in Nature Communication unless significant changes are made. The authors are advised to address the following comments carefully.	We thank the reviewer for constructive feedback and provide detailed responses below where we indicate changes made.
A. Formatting 1. The article should be written with clear and structured sections and subsections.	We followed Nature Communications guidelines on section headings ("Introduction of referenced text that expands on the background of the work (some overlap with the abstract is acceptable), followed by sections headed Results, Discussion (if appropriate) and Methods (if appropriate [...]). The Results and Methods sections should be divided by topical subheadings", see https://www.nature.com/ncomms/submit/article).
2. The references are written manually; some texts need references but are missing. It is recommended that the authors use referencing software to assist in article writing.	We have in fact used Zotero for reference management.
3. There are a total of 69 references in the text (body text and methodology), but in the bibliography, there are only 56 references. The authors should resolve this problem.	We did however not include a reference list in the Data and Methods part. We have fixed this problem.
4. Most cited academic papers are from the period 2015-2018. It may be more useful if the authors could add more recent references, such as from 2021; the dynamics of the energy sector (renewable energy (RE) vs fossil fuels) have developed more rapidly in recent years.	We have now added more recent references to the manuscript, including a few that concern the renewable versus fossil fuel dynamics. However, we find that among more conceptual academics papers addressing this dynamic, the insights of the older (2013-2018) papers are still relevant.

REVIEWER COMMENTS

	This is for instance the case for Cherp et al (2018) and Newell & Johnstone (2018).
B. Substance and Body of Analysis 1. If the study focuses on OECD countries, the title should be revised to state it.	Thank you, this is highly useful advice. We have revised the title to include “in OECD countries”.
2. P2 L77-78: “Policy interventions supporting RE helps to transition the economy away from fossil fuel.” In this citation, policy intervention to RE includes subsidy as well. How could the authors rationalize or differentiate the subsidy for RE and subsidy for fossil fuels in the model?	We agree that the relationship between RE and fossil fuel subsidies is interesting. Yet, given that we focus on fossil fuel subsidies in our analysis, we are interested in how the RE sector itself, rather than policy interventions to promote RE, influence fossil fuel subsidies and their reform. Including RE subsidies in our model would complicate matters. The relationship between RE subsidies and fossil fuel subsidies is a topic for future research.
3. P3 L87-89: “Subsidies provide concentrated benefits for specific interest groups that produce or consume fossil fuels, subsequently increasing their share of the economy.” Fossil fuel subsidy in the Middle East and some Asian countries (such as Indonesia) characterizes universal subsidy, likewise, bioethanol subsidy in developed countries (such as Australia). How could the author differentiate these 2 types of subsidy in the model?	First, among the OECD countries studied here, a large share of the subsidies, probably more than half, are not universal in scope but are limited in terms of targeting specific parts of the population (e.g. pensioners) or sectors (e.g. tax exemptions for fuel used by agriculture or transfers to coal mines). Second, and more importantly, despite the universal character of some subsidies, many studies, including those cited within the article (e.g. Victor 2009; Arze del Granado 2012; Blankenship & Urpelainen 2019; Fails 2022) highlight the unequal benefits they provide. In particular, wealthier segments of the population generally receive greater benefits, being more likely to be motorists or own and use multiple (larger) vehicles (Arze del Granado 2012). Similarly, we consider motorist advocacy groups or industries that are more suited to, or dependent on, using these fuels intensively to be receiving a concentrated benefit over those where fuel costs have less impact. For example, the steel industry is likely to receive greater benefit from a coal subsidy than the software industry even if the subsidy is universal. In terms of differentiation, we do not believe this is particularly useful for the arguments or the model within the context of the paper. The information has been collected within the

REVIEWER COMMENTS

	dataset but does not impact the model due to the “concentrated benefit” idea outlined above.
4. Figure 2 P3 describes the structural causal model of how the energy market and political institutions interact when it comes to fossil fuel subsidy reforms. There is a flaw in the concept of this model; the energy market is not the same as (has a larger scope than) the renewable energy market. Hence, this model needs to be revised to be more representative; what subsystems of the energy market are involved in this model?	This and the following comments are really helpful and point out an important need for a clearer figure and underlying concepts. Some inconsistencies and imprecisions are related to how we operationalize variables, but also how we represent them in the causal model in Figure 2. Regarding energy market and renewables, we use data on “Renewable energy consumption (% of total final energy consumption)” from the World Bank’s World Development Indicator Database as explained in Data and methods line 161. That means we do recognize that the energy market is larger and more encompassing than the renewable energy market and measure the share of renewable energy consumption in the energy market. We do not subset to the renewable energy market.
5. Figure 2 P3: It is unclear what basis or approach the authors used to build the causal model. This should be explained and is important to be explained in the methodology as well, and how this causal model is combined with the regression and random forest analysis?	The information on how we build our causal model is explained in the Methods section from line 455 (p.13). We proceeded through an inductive analysis, where we let the data speak for itself by identifying variables with high explanatory power through the use of regression trees (lines 458 ff.). Information on the deductive model testing through two-way fixed effects is provided in lines 488 ff. However, we do agree that we could have made the approach more transparent and have thus provided corresponding information in the introductory paragraph in the Data and Method section.
6. Figure 2 P3: What do the rectangle and circle symbols mean? If we refer to the system dynamics concept, both symbols represent different things.	This and the following comments are really important. We do acknowledge that our representation was not entirely consistent and have worked on clarifying the figure. We take inspiration from causal structural models, structural equation modelling but not really system dynamics symbolism. We formulate / hypothesize causal relations (symbolized by arrows) between variables (symbolized by ovals). The former presentation included a feedback mechanism as if it was a variable, which it is not.

REVIEWER COMMENTS

	This is important to correct, and we thank the reviewer for making us aware of this.
7. Figure 2 P3: Based on the concept of causal modeling, a variable will influence another variable not the flow (in the Figure, flow is represented by an arrow sign). Another problem, there is an inconsistency about the arrow from “fossil fuel subsidy levels” to or from “lock-in effects.” The authors should explain the rationale for these model-building problems. It cannot be seen clearly how the flow of this figure is translated into the regression model. Meanwhile, each symbol has some proxies, such as the quality of political institutions measured by government effectiveness and corruption level? Figure 2 really needs to be revised.	We would like to point out that there are causal loop diagrams that come with their own symbolism. Given that we take inspiration rather from Directed Acyclic Graphs used in structural causal modelling than from system dynamics, arrows do represent causal relations rather than flows. We do not strictly adhere to an acyclic model formulation as we think there is an important feedback mechanism at play. Thus we have formulated a theory were we stipulate that the quality of institutions and the fossil fuel subsidy levels influence the causal relation between other variables. This is indicated by arrows pointing at arrows. It is in line with how moderation models are generally drawn (see e.g. Wu, A. D., & Zumbo, B. D. (2008). Understanding and using mediators and moderators. Social Indicators Research, 87, 367-392.). We do, however, recognize that we should have made this clearer and have thus revised the accompanying text too. To explain the structure and mechanisms more clearly, see lines main document 130 ff. It is true that we operationalize the quality of institutions concept with multiple variables, and we indicate that more clearly now in the figure caption.
8. If the causal model as Figure 2 P3 refers to a causal loop diagram (system dynamics), the logic and model building need to be revised	We use the logic of structural causal models and moderating effects. But we agree that the clarity of explaining our model needed improvements (see comment above, too).
9. Unclear cross-references or links between the causal model, hypothesis, variables, and measurement, and the regression model. The authors need to provide informative cross-references among these aspects.	It is not quite clear here to us what is asked for. We have worked on explaining the relation between hypotheses and the causal theoretical model better (page 4 ff) and we have also provided more references to the literature in line 179ff. We assume that the comment mainly refers to the following points 10-14.
10. P4 L125-131: “Our theory consists of three connected mechanisms: i) a market-power mechanism that captures the effects of competition	Yes, we agree that this needs to be clear.

REVIEWER COMMENTS

between energy sectors, and where larger renewable energy shares decrease the role of fossil fuels and thereby ease the reform of fossil fuel subsidies; ii) a policy mechanism where reforms lead to lower overall fossil fuel subsidy levels; iii) a feedback mechanism that captures the effects of institutional quality (e.g., government effectiveness, corruption control), lock-in into fossil fuel-dependent pathways, and their subsequent effect on the market-power mechanism as well as the policy mechanism.” The authors need to confirm clearly what the market-power mechanism, policy mechanism, institutional quality represent? It is unclear if the variables government effectiveness and corruption control are only the examples of institutional quality, so what about many other variables or factors?	 1) We have now improved on the link between the causal theoretical model and the text (lines 130 to 143) 2) The mechanisms are explained in subsequent sections. 3) The operationalization of variables is explained in the data and method section where we systematically present the two outcome variables fossil fuel subsidy reforms and fossil fuel subsidy levels (subsubsection outcome variables) and independent variables (subsubsection Set of explanatory variables). There we indicate the data sources of the different variables and their measurements. We have clarified the relation to concepts in the 4) For details on variable selection through regression trees and precise formulations of regressions for the deductive theory testing, we refer the reviewer to the Abductive analysis subsection of the Data and Method section. 5) We have however also improved on the presentation of results, where we explain which variables have been used in the
11. Hypothesis 1: the larger the market share of alternative technologies, such as renewables, the easier it is to reform fossil fuel subsidies. Hypothesis 1 has weakness and is not straightforward; there are lots of alternative technology types, meanwhile, in this hypothesis, renewables are only the example? This hypothesis also inconsistent with the causal model built; the renewables are the focus of investigation (not an example).	We agree that there are many alternative technologies. We do believe that the mechanism might also hold for other technologies but we have not tested that. We have operationalized the hypothesis it with a “renewable energy share of total energy consumption” variable (see line 394 in Data and Methods). To be more accurate we have formulated the hypothesis more crisply now with regard to what we actually test.
12. P5 L161-162: The third type of mechanism is placed on the system level and captures how structural aspects of the socio-economic system influence the functioning of the main causal chain. What is the main causal chain in this statement? Is there any supporting causal chain?	We agree that this was not quite clearly formulated. We have made a clearer reference to what we understand as the main mechanisms in line 175f.
13. Hypothesis H1, H2, H3.1, H3.2, and H4 need to be revised/adjusted once comments numbers B9, 10, and C6, 8.	We thank the reviewer for the comment and have made minor clarifications such that the hypothesis more accurately reflect what we

REVIEWER COMMENTS

	actually test. We also split the mechanisms into four and reorganized the presentation. We hope this helped to clarify.
14. H3.2 seems to have an intermediary variable, i.e., the level of fossil fuel subsidy influences the renewables' share via lock-in effects. How the authors could reveal and explain this intermediary calculation?	We estimate an interaction effect. See both the revised text in line 185f. in the main manuscript as well as details in Data and Methods line 511.
15. P7 L212-216, the sentences in these lines are unclear, vague, inconsistent, and contradictory between each other; needs to be revised. References are needed as well.	Given that there was also another reviewer's comment on the lack of explanation of interaction effects we have also tried to make the explanation in the main text more clear (eg. lines 211 ff. and lines 243ff).
16. The Discussion and Conclusion Section and Abstract need to be revised after all issues highlighted have been resolved. Here, the authors also need to summarize: i) how the methodology contributes to answering the problems being investigated, ii) the answer of hypotheses being tested.	We have revised the discussion and conclusion section and included pointers to the methodology and more clearly discuss findings in relation to hypotheses.
C. Data and Methods 1. Unclear political economy theory approaches used and referred to in this paper (both in body analysis and methods).	We agree that the term "political economy" is rather broad and has taken on many different meanings. To clarify what we mean, we have added the following sentence: "Our political economy approach shall be understood as one that studies political and economic institutions and actors, as well as their interaction".
2. P1 L1-2: "To explore and assess the relationship between both i) the number of fossil fuel subsidy reforms and ii) the amount of fossil fuel subsidies, we make use of regression trees and random forests." The logic of the random forest cannot be seen in the model or calculation.	We have revised the introductory paragraph to explain that we used bagged regression trees for an inductive exploration of variable importance to inform the building of theory – which is then again tested against data. This process is called abductive and we have tried to make this clearer in the method description now (line 283ff).
3. Sentences on P1 L21-25 seem contradictory and unclear. The authors need to specify what is the proxy or measure that they use and differentiate for electricity subsidy or fuel subsidy? Are both subsidies used in the modeling? What is the judgment of 60% constraint in the subsidy reform?	We have revised these lines to rationalize our choice of the 60% constraint. We have also clarified how we differentiate between electricity subsidies and fossil fuel subsidies and provide some examples (lines 333ff.)
4. What is the judgment of classifying the constraints of the percentage of subsidy from the national budget as P1 L40-48? The sentence writing is also unclear; needs to be rewritten. Such as: if the value of a subsidy is at or above 1% of the national budget, the	We have added an explanation of our reasoning for the inclusion criteria and attempted to provide a clearer justification for these constraints. We have also added examples to

REVIEWER COMMENTS

reform must represent a 5% change in the value of that subsidy. Still unclear about the relationship and judgment about 1% national budget and 5% change of subsidy value, and the reform itself. Likewise for the subsequent points.	illustrate why these inclusion criteria are necessary (lines 353 ff).
Meanwhile, in P2 L49 stated: These rules (L40-48) cannot be applied to increases in fixed fuel prices. Are the authors able to provide relevant explanations and reasons about this matter? What does “increases in fixed fuel prices” mean? On the other hand, the nature of fuel is fluctuating.	We have expanded our explanation and added some clarifications that we hope make things a bit clearer and justify the choices we made (lines 375 ff).
5. How the authors could explain criteria as stated in P2 L73-92 and Table 1 are aligned with the variables in Table 2. What are the relationships between the criteria and variables?	We have more information in the fossil fuel subsidy data than what we finally make use of when estimating regression models. As we estimate three-way interaction models which are already complex, we have refrained from additionally conditioning on the type of fossil fuel subsidy reform. We only estimate average effects of FFS reforms across all types of reform. Thus, any reform is operationalized in the variable “Number of reforms” in Table 1. We leave more detailed analysis for future research.
6. What is the unit of measurement in Table 1?	The units in Table 1 are the number reforms per year, we have specified this in the table caption now.
7. The authors need to explain the rationality of selected or chosen variables.	We conduct a two-step analysis. First the regression trees tell us which variables are generally important (see Data and Methods subsection on Inductive Analysis and Supplementary Material). Then we test derived hypothesis deductively. Here have explained the variable selection procedures in lines 502 f.
8. The authors need to explain briefly the validation process of modeling and the calculation.	The reviewer can find information on cross-validation of bagged regression trees in lines 471 ff in Data and Methods, and on the robustness of the two-way fixed effects estimation in lines 506ff.
Reviewer #3:	
The authors propose an analysis of fossil fuel subsidy reforms developed under the lens of political economy theory. They employ an abductive approach to identify the relevant variables and develop a theory	We thank the reviewer for the very important and helpful feedback and provide our responses below.

REVIEWER COMMENTS

about causal mechanisms based on existing literature. Then the hypotheses derived from the theory are empirically tested on a new dataset, mapping fossil fuel subsidy reforms from 35 OECD countries from 2005-2020. More specifically, they investigate three mechanisms: i) a market-power mechanism, according to which larger renewable energy shares ease the reform of fossil fuel subsidies (RH1: larger renewable energy shares, higher number of reforms);ii) a policy mechanism, according to which reforms lead to lower overall fossil fuel subsidy levels (RH2: higher number of reforms, lower level of subsidies);iii) a feedback mechanism, according to which mechanisms i) and ii) are affected by institutional quality (i.e., positively influencing both the positive correlation in RH1 (RH3.1), and the negative correlation in RH1 (RH4)) and path dependence (i.e., negatively influencing the positive correlation in RH1 (RH3.2)). The model is estimated by means of panel fixed effect model. A preliminary inductive exploratory analysis is run through a regression tree, multiple visual inspections with a random variation, and finally a tree bagging approach to derive variable importance.	
My first concern is related to how the data selection and the construction of the database may drive the results. This is because the dataset on the number of reforms only includes those interventions that remove or reduce support for fossil fuels, conditional on additional inclusion constraints (e.g., "If the value of a subsidy is at or above 1% of the national budget the reform must represent a 5% change in value of that subsidy") as detailed at pag. 1 in the Supp. Material. In this respect, the authors should make a larger effort to explicitly discuss their selection criteria, the motivation behind and possible bias determined by these choices (particularly with respect to the hypothesis tested and endogeneity concerns in the empirical analysis). In the current form, the paper seems more about fossil fuel subsidy removal or reduction, rather than the more general idea of fossil fuel subsidy reform. In addition, it is not clear from	This feedback is helpful, and we hope we're a bit more transparent on the data collection process. We have expanded our explanation of the inclusion criteria and provided examples to illustrate why we believe these are necessary. We have also added a section discussing the process by which the dataset was constructed, some considerations of its issues and how we have attempted to mitigate these issues.

REVIEWER COMMENTS

which sources information have been collected, and I believe this is an important aspect since the authors emphasise the novelty of the dataset.	
My second main comment is related to the interpretation of the results, especially with respect to the evidence provided for testing the feedback mechanisms, operationalised through the introduction of different interaction terms in the model specifications. The interpretation of marginal effects is not trivial when introducing different interactions, but the paper does not provide the reader with any guidance. The paper would largely benefit from discussing the estimated coefficient of the interactions (and their statistical significance), indicating why (or why not) results are consistent with theory.	That point is well received. We have made an effort to better link the explanations between figures, text and pointers to the regression results. The reviewer will also find that we reorganized the different mechanisms a bit and thus hopefully made the overall presentation a bit clearer. Specifically, we modified and hopefully clarified the explanation for more complex interaction models (page 6 lines 211 ff.).
More generally, the authors should make a larger effort to discuss their estimations and explain their correspondence with the research questions addressed in the paper. For instance, in model 3 table A2 there are three interaction terms (i.e., renewable_share:total_subsidies_cap, renewable_share:government_effectiveness, total_subsidies_cap:government_effectiveness) and only the third one is statistically significant. The authors suggest that their findings support RH3.2, a conditional support for RH3.1 and mixed evidence for RH4. Unfortunately, the paper does not provide any explanation for these results.	This was an important pointer. We do now systematically go through the models and point to the supporting evidence from the regression tables and significance levels in the main text. The reviewers comments in fact led us to revise the results regarding hypothesis (now H4 formerly H3.2) as the interaction effect was not significant albeit the predicted functional forms seems to indicate a considerable effect. We are thus grateful to the reviewer for spotting this.
In addition, few minor points: - No reference list in the supplementary materials	Yes, this was an unfortunate oversight. We have now added the reference list to the Data and Methods section. The Appendix did have a reference list.
- It is not clear the time span covered: the new dataset (should) cover the period from 2005 to 2020, but different time period are reported in different figures (e.g., 2000-2020 in figure 1; 2010-2019 in figures 3-4).	The times covered in different plots are indeed not entirely consistent, but a result of different data availabilities. To not throw away data, we have maximized each corresponding data in which variables were available. However, we did indeed spot that Figure 1 is only covering 2005-2020, and Figure 3-5 indeed 2010-2019

REVIEWER COMMENTS

Reviewer #1 (Remarks to the Author):

We thank the author(s) for addressing all points during the review process. While I am generally still not convinced about the overall methodological approach, I think that the authors are now much more explicit and transparent about the shortcoming of their work.

Reviewer #1 (Remarks on code availability):

While I have not run a full replication analysis, the provided code and data seem to enable a replication. Not everything is clearly documented but I think that all important elements for a replication are available.

Reviewer #2 (Remarks to the Author):

The authors have addressed most of my comments and made relevant revisions.

Reviewer #3 (Remarks to the Author):

I appreciate how the authors have improved the description of their model. Yet, I still have some concerns.

First, the authors clarified that their dataset only includes 'positive' reforms, i.e., those reforms which remove or reduce support for fossil fuels, however I still believe that the authors should make a further in discussing the trade-off of this selection criteria, on the interpretation of the results and in terms of implication for causality. For instance, what if other reforms, excluded from the dataset, are likely to over-balance the effect of the selected reforms? Similarly, at lines 396 ff "For example, a tax break applied to natural gas that has been renewed yearly for 5 years before being allowed to expire would be included; one that was introduced for a five-year period and then allowed to expire would not be". The authors should further comment on the possible limitation due to the selection.

Second, my concerns related to the interpretation of the results has not been fully answered. These concerns apply especially to H3.2.

H3.2 is defined as follows: "The negative correlation between fossil fuel subsidy reforms and fossil fuel subsidy levels is positively influenced by institutional quality in terms of government effectiveness and corruption control." Here, the institutional quality is measured by means of two alternative variables 'government effectiveness' and 'corruption control'. Hence, I expected H3.2 to be analysed by looking at two interaction terms between the number of fossil fuel subsidy reforms ('total number reforms' in table 2) and either 'government effectiveness' or 'corruption control' (observe that among these two-way interaction terms, only one, i.e., total_number_reforms:government_effectiveness, is statistically significant). However, the authors

refer to a three-way interaction terms

(total_number_reforms:government_effectiveness:corruption_control) when discussing this hypothesis. It is not clear why, thus it should be not clearly explained and needs further discussion.

To further stress their argument about the heterogeneity of the results depending on the institutional quality, the authors could provide the quantification of the net marginal effects, e.g., the marginal effect associated to the total number reforms at different of government effectiveness, and test whether these are statistically different.

More in general, these issues overall lead to some causality concerns in the interpretation of the results. For instance, according to H1, larger renewable energy consumption determines larger number of reforms (i.e., larger number of reforms to reduce/remove fossil fuels subsidies), but it can also be case that because of larger number of reforms to reduce/remove fossil fuels subsidies, fossil fuel consumption becomes relatively less competitive and as a result consumption of renewable energy increases. Hence, to claim causality, a larger effort is required to strengthen the empirical analysis or the theoretical contribution. On this latter point, the authors may want to better discuss their theoretical approach, with reference to existing studies. For instance, it may be appropriate to add references at about lines 37-38 (Our political economy approach shall be understood as one that studies political and economic institutions and actors, as well as their interaction”) and lines 126 ff (“This theory was developed by placing the above-mentioned factors in a wider theoretical context”).

Minor:

Given that in the revised version of the paper greater emphasis has been given to the country selection with respect to previous studies (OECD vs. developing countries), the paper would benefit from a comparison of their results with respect to existing literature with the aim of highlights differences/similarities.

The authors should check the writings as there are some typos to be corrected. E.g.:

- Line 213: H3.2 instead of R4?
- Check if the arrows in figure 2 correctly represent the RHs
- Lines 240ff: “[...] provide indications that that the effectiveness [...]”
- Lines 321ff: “Through each of these steps we relevant legislation, government [...]”
- Lines 440ff: “in particular = “Oil Production – Barrels” [...]”

Reviewer #3 (Remarks on code availability):

No code provided.

REVIEWER COMMENTS

Reviewer 3 (Remarks to the Author):	Response
I appreciate how the authors have improved the description of their model. Yet, I still have some concerns.	We thank the reviewer for the constructive and helpful feedback. We addressed the concerns raised and respond to them in detail here and point to corresponding changes in the manuscript.
First, the authors clarified that their dataset only includes ‘positive’ reforms, i.e., those reforms which remove or reduce support for fossil fuels, however I still believe that the authors should make a further in discussing the trade-off of this selection criteria, on the interpretation of the results and in terms of implication for causality. For instance, what if other reforms, excluded from the dataset, are likely to over-balance the effect of the selected reforms? Similarly, at lines 396 ff “For example, a tax break applied to natural gas that has been renewed yearly for 5 years before being allowed to expire would be included; one that was introduced for a five-year period and then allowed to expire would not be”. The authors should further comment on the possible limitation due to the selection.	We have attempted to address this comment through a rewrite of the sections highlighted by the reviewer. On the first point, we have clarified the choices we have made regarding inclusion, discussed limitations of the dataset and where it is appropriate to use and how we keep within these limitations for our tests in the first paragraph discussing the dataset in the methods section (lines 322-345). As we are interested in the effects of meaningful reforms that are meant to reduce fossil fuel subsidies, we needed to be conservative in including reforms in the dataset. We have been careful not to include reforms that were e.g. rolled back quickly as that would possibly let us to underestimate the effect on levels. To increase transparency, we have expanded our reasoning on why the selection criteria is made and why it is appropriate for the paper. We have also discussed the potential limitations of the selection criteria and attempted to justify our choices (lines 446-452). It is important to note that while reforms have only been included if they reduce or remove fossil fuel subsidies, subsidy levels are taken from a separate dataset. In cases where we see increasing subsidy levels per capita (see lefthand part of figure 6) following reform, we argue that the reform process is being captured and the reforms we see are in fact being overbalanced by excluded reforms or the introduction of new subsidies. We see, however, that this only happens under certain conditions (high gov effectiveness, low corruption control). We hope this addresses points well made by the reviewer.
Second, my concerns related to the interpretation of the results has not been fully answered. These concerns apply especially to H3.2. H3.2 is defined as follows: “The negative correlation between fossil fuel subsidy reforms and fossil fuel	We started the analysis thinking that both variables could matter and wanted to see which it was. The three-way interaction was included out of curiosity. Somewhat unexpectedly, we found that the effect are conditional on one another – or rather the positive interaction effect of

REVIEWER COMMENTS

subsidy levels is positively influenced by institutional quality in terms of government effectiveness and corruption control.” Here, the institutional quality is measures by means of two alternative variables ‘government effectiveness’ and ‘corruption control’. Hence, I expected H3.2 to be analysed by looking at two interaction terms between the number of fossil fuel subsidy reforms (‘total number reforms’ in table 2) and either ‘government effectiveness’ or ‘corruption control’ (observe that among these two-way interaction terms, only one, i.e., total_number_reforms:government_effectiveness, is statistically significant). However, the authors refer to a three-way interaction terms (total_number_reforms:government_effectiveness:corruption_control) when discussing this hypothesis. It is not clear why, thus it should be not clearly explained and needs further discussion. To further stress their argument about the heterogeneity of the results depending on the institutional quality, the authors could provide the quantification of the net marginal effects, e.g., the marginal effect associated to the total number reforms at different of government effectiveness, and test whether these are statistically different.	government effectiveness depends on corruption control. We have clarified this in the text, mentioning both the two-way interactions and their significance as well as explaining the motivation to include a three-way interaction (see lines 238 ff). Note that we in Hypothesis 3.1 have changed the variable from “government effectiveness” to “electoral democracy” (see elaboration of this change below). What we have done in the paper is to provide the predictions of the interaction effects, ceteris paribus. Here, the other variables are held constant. In our view that already substantiates effect heterogeneity. However, we have now also computed partial derivatives / marginal effects for total number of reforms on a grid for different levels (i.e. quantiles) of both government effectiveness and corruption control. We have included the resulting table as an Excel file into the supplementary material and commented on it in the Method section (lines 563). Not all marginal effects are significant but 10 out of 25 for the computed grid, with a tendency for significance at higher government effectiveness.
More in general, these issues overall lead to some causality concerns in the interpretation of the results. For instance, according to H1, larger renewable energy consumption determines larger number of reforms (i.e., larger number of reforms to reduce/remove fossil fuels subsidies), but it can also be case that because of larger number of reforms to reduce/remove fossil fuels subsidies, fossil fuel consumption becomes relatively less competitive and as a result consumption of renewable energy increases. Hence, to claim causality, a larger effort is required to strengthen the empirical analysis or the theoretical contribution.	We partially agree on this point. Fossil fuel subsidy reforms can indeed change the competition between technologies and tilt the market towards favoring renewables. However, we would think that it is not directly the reform that changes the competition but rather the subsidy levels in the energy market. So we would expect the causal chain would be as follows: reforms reduce subsidies, and subsidies change competition. We have thus modified both our theoretical causal model, i.e. Hypothesis 4 as well as the corresponding estimation to see if fossil fuel subsidy levels impact renewables market share. The corresponding coefficient is negative and small but significant at 10% level with clustered standard error á la Driscoll and Kraay. This however also meant that we needed to re-estimate the model for Hypothesis 3.1 because it should no longer include subsidy levels. As before, government effectiveness was not significant in the interaction model. Based on our two-directional selection approach for the explanatory variables the term was correspondingly dropped. We have therefore

REVIEWER COMMENTS

	now re-estimated model with the electoral democracy index, which remained significant (again, the same as before). We have now included electoral democracy in the interaction term and find it to be significant.
On this latter point, the authors may want to better discuss their theoretical approach, with reference to existing studies. For instance, it may be appropriate to add references at about lines 37-38 (Our political economy approach shall be understood as one that studies political and economic institutions and actors, as well as their interaction”) and lines 126 ff (“This theory was developed by placing the above-mentioned factors in a wider theoretical context”).	We expanded on our theoretical approach and its roots in the studies of the political economy of energy in the two sections mentioned. Particularly, we elaborated on how we see energy politics, including fossil fuel subsidies, as shaped by actors from respectively the fossil fuel and renewable energy constituencies, and the constraints they face in terms of domestic political institutions.
Minor: Given that in the revised version of the paper greater emphasis has been given to the country selection with respect to previous studies (OECD vs. developing countries), the paper would benefit from a comparison of their results with respect to existing literature with the aim of highlights differences/similarities.	Thank you for this useful suggestion, we have added a section to the Discussion that addresses this issue (lines 286ff.).
The authors should check the writings as there are some typos to be corrected. E.g.:  - Line 213: H3.2 instead of R4? - Check if the arrows in figure 2 correctly represent the RHs - Lines 240ff: “[...] provide indications that that the effectiveness [...]” - Lines 321ff: “Through each of these steps we relevant legislation, government [...]” - Lines 440ff: “in particular = “Oil Production – Barrels” [...]” 	Thank you, this was helpful!
(Remarks on code availability): No code provided.	Yes, we have in fact provided code, as well as data but we only indicated this in the submission system not in the manuscript. We have fixed this now. Here is the link: https://zenodo.org/doi/10.5281/zenodo.10320470

REVIEWER COMMENTS

Reviewer #3 (Remarks to the Author):

The authors have answered to most of my comments and implemented pertinent revisions. Although I still have few reservations regarding the overall methodological approach, I acknowledge that the authors have significantly enhanced the transparency and clarity regarding the limitations of their study.

Reviewer #3 (Remarks on code availability):

This is not a push the button replication file.

As such, when trying to execute the code several errors are returned. My suggestion is to check the code again and make sure that the script can be run smoothly on every platform. Only under this condition the code can be examined.

On a related matter, I also invite the author(s) to check the excel file 'MarginalEffects.xlsx', as the only data included seems associated to the country Australia for year 2010.

REVIEWER COMMENTS

Reviewer #3 (Remarks to the Author):	Response
The authors have answered to most of my comments and implemented pertinent revisions. Although I still have few reservations regarding the overall methodological approach, I acknowledge that the authors have significantly enhanced the transparency and clarity regarding the limitations of their study.	We thank the review for helping us making the analysis more transparent and double checking our results.
This is not a push the button replication file. As such, when trying to execute the code several errors are returned. My suggestion is to check the code again and make sure that the script can be run smoothly on every platform. Only under this condition the code can be examined.	It is not a push the button replication file. The R programming language needs to be installed and quite a few packages. We have now checked the code again, and facilitated an easier replication by:  - Improving the readme file to orient the interested researcher on how to replicate our results (Readme.txt) - Checking if packages are installed and if not installing them (at the beginning of each R script) - Reading out the location of the folder where the replication files are being placed and using that as a working directory (within each R script) - Double checking that relative file paths all make sense and work - Also providing code for the analysis in the supplementary materials, i.e. the regression trees in script 05_supplementary_abduction.R To test this, we have run the whole analysis on a different computer without the packages installed and with the downloaded zip file from the Zenodo repository being placed in a random location. Thereby we hope to make this as easily replicable as possible. We have not tested it on other platforms than Windows, but running it should in principle not create any problems as long as R is installed.
On a related matter, I also invite the author(s) to check the excel file 'MarginalEffects.xlsx', as the only data included seems associated to the country Australia for year 2010.	Regarding the MarginalEffects.xlsx, we have now provided the reviewer with additional marginal effects by varying also countries (all countries) and years (2010, 2015, 2019).